# Inner hair cell dysfunction in *Klhl18* mutant mice leads to low frequency progressive hearing loss

**Neil J. Ingham** *, **Navid Banafshe, Clarisse Panganiban, Julia L. Crunden, Jing Chen, Morag A. Lewis, Karen P. Steel**

Wolfson Centre for Age-Related Diseases, King's College London, London, United Kingdom

* neil.ingham@kcl.ac.uk

**Data Availability Statement:** The data underlying this study are available on the Dryad repository (DOI: 10.5061/dryad.3tx95x6h4).

## Abstract

Age-related hearing loss in humans (presbycusis) typically involves impairment of high frequency sensitivity before becoming progressively more severe at lower frequencies. Pathologies initially affecting lower frequency regions of hearing are less common. Here we describe a progressive, predominantly low-frequency recessive hearing impairment in two mutant mouse lines carrying different mutant alleles of the *Klhl18* gene: a spontaneous missense mutation (*Klhl18^lowf*) and a targeted mutation (*Klhl18^tm1a(KOMP)Wtsi*). Both males and females were studied, and the two mutant lines showed similar phenotypes. Threshold for auditory brainstem responses (ABR; a measure of auditory nerve and brainstem neural activity) were normal at 3 weeks old but showed progressive increases from 4 weeks onwards. In contrast, distortion product otoacoustic emission (DPOAE) sensitivity and amplitudes (a reflection of cochlear outer hair cell function) remained normal in mutants. Electrophysiological recordings from the round window of *Klhl18^lowf* mutants at 6 weeks old revealed 1) raised compound action potential thresholds that were similar to ABR thresholds, 2) cochlear microphonic potentials that were normal compared with wildtype and heterozygous control mice and 3) summating potentials that were reduced in amplitude compared to control mice. Scanning electron microscopy showed that *Klhl18^lowf* mutant mice had abnormally tapering of the tips of inner hair cell stereocilia in the apical half of the cochlea while their synapses appeared normal. These results suggest that Klhl18 is necessary to maintain inner hair cell stereocilia and normal inner hair cell function at low frequencies.

## Introduction

Progressive hearing loss with age is the most common sensory deficit in the human population and it can begin at any age. Genes found to be involved in hearing loss can give valuable insights into the molecular pathways and pathological processes leading to progressive deafness. Mouse mutants can not only provide valuable candidate genes for human deafness but also reveal novel pathological mechanisms underlying hearing loss. We have used the mouse

**Funding:** This work was supported by the Wellcome Trust (098051, 100699, and WT089622MA; to KPS), the Medical Research Council (G0300212 to KPS), Royal National Institute for Deaf People (to KPS and NJI) and the EC (MSCA-ITN-2016-LISTEN-722098 to KPS). The funders had no role in study design, data collection and analysis, decision to publish, or preparation of the manuscript. For the purpose of Open Access, the author has applied a CC BY public copyright licence to any Author Accepted Manuscript version arising from this submission.

**Competing interests:** The authors have declared that no competing interests exist.

to identify multiple genes involved in hearing loss but it is clear that many more genes, and pathological mechanisms, remain to be discovered before we have a full understanding of the disease [1].

In this report we explore the role of the *Klhl18* (Kelch-like family member 18) gene in hearing loss using two different mutations in the mouse. The first allele (termed *Klhl18^{lowf}*) arose as a spontaneous missense mutation (Val55Phe) predicted to have a damaging effect on protein structure [2]. These mutants have normal middle ears and no gross malformations of the inner ear were found [2]. The second allele is a targeted mutation, *Klhl18^{tm1a(KOMP)Wtsi}*, exhibiting low frequency hearing loss in adult mice [1,3]. Mice carrying the targeted allele have been subjected to a broad-spectrum phenotyping screen as described by White et al. [4]. The only other significant phenotype found was a decreased volume and thickness of femur trabecular bone in female homozygous mutants [1]. Neither allele produced overt vestibular phenotypes such as circling or head-bobbing and homozygotes are viable and fertile [1,2]. The hearing loss is inherited in a recessive manner.

Kelch-like family member 18 is part of a 42-member superfamily of genes [5]. In the mouse, Klhl18 is a 574 amino acid protein containing a Bric-a-Brac, Tramtrack and Broad complex (BTB) domain, a BACK domain, and six Kelch β-propeller domain repeats which typically have roles in extracellular communication, cell morphology, and actin binding [5]. BTB domains are involved in protein-protein binding [6,7] and have been associated with a variety of cellular mechanisms, including cytoskeletal organization [8], voltage-gated potassium channel opening [9], transcriptional regulation [10] and targeting of proteins for ubiquitination [11,12]. *Klhl18* encodes an adaptor protein for the Cul3 ubiquitin ligase, providing specific targeting of Aurora-A for ubiquitination and subsequent initiation of mitotic entry [13]. Recently, Chaya and colleagues [14] have implicated *Klhl18* in retinal photoreceptor function in mice through targeted ubiquitination of Unc119.

Here, we describe the progressive elevation of auditory brainstem response (ABR) thresholds in *Klhl18* mutant mice from 4 weeks old. In contrast, *in vivo* measurements of distortion product otoacoustic emissions (DPOAEs) and cochlear microphonics (CM) indicated normal outer hair cell (OHC) function is maintained. The number of synaptic contacts between inner hair cells (IHCs) and afferent cochlear neurons was normal. However, IHC stereocilia displayed abnormal lengthening and tapering of their distal ends, especially affecting the apical half of the cochlear duct. These observations indicated the IHC as the primary site of the pathology. Summating potentials (SP, a sustained dc shift in voltage seen during sound exposure and thought to arise mostly from depolarisation of IHCs) were abnormally small in homozygous mutants, supporting this suggestion.

## Materials and methods

### Ethics statement

Mouse studies were carried out in accordance with UK Home Office regulations and the UK Animals (Scientific Procedures) Act of 1986 (ASPA) under UK Home Office licences, and the study was approved by the King's College London and Wellcome Trust Sanger Institute Ethical Review Committees. Mice were culled using methods approved under these licences to minimize any possibility of suffering.

### Mice

The mouse lines carrying the two alleles of *Klhl18* used in this study originated from the Wellcome Sanger Institute Mouse Genetics Project, both generated and maintained on a C57BL/6N genetic background. One was a spontaneous missense mutation, *Klhl18^{lowf}* (Fig 1A), that

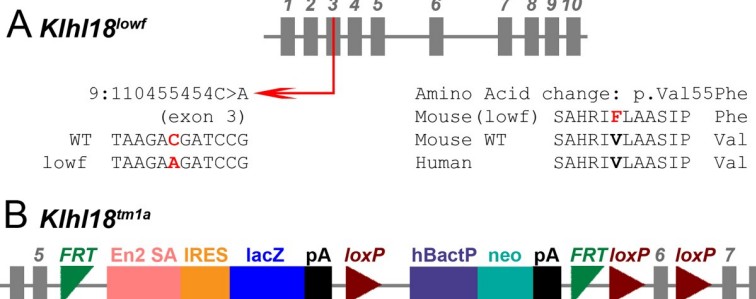

**Fig 1. Schematic representations of the *Klhl18^lowf* and *Klhl18^tm1a* alleles. A.** *Klhl18^lowf* is formed by a single base pair change on Chr9, at position 110455454, in exon 3 of the gene, with the wildtype C replaced with the mutant A. This produces an amino acid substitution in the resultant protein with the wildtype valine (conserved across species) being replaced by phenylalanine at position 55 (see [2] for further information). **B.** *Klhl18^tm1a* was generated at the Wellcome Sanger Institute by insertion of a large DNA cassette before the critical exon (exon 6) of the gene, which interrupts transcription of the gene into mRNA. The cassette is composed of a number of components; En2 SA (engrailed2 splice acceptor), IRES (internal ribosome entry site), lacZ (beta-galactosidase reporter gene), pA (polyadenylation site), hBactP (beta-actin promotor part) and neo (neomycin resistance gene), which are flanked by FRT (Flp-FRT) and loxP recombination sites to produce the promoter-driven Knockout First, reporter-tagged insertion with conditional potential allele (tm1a) of *Klhl18* (see [15] for further information).

occurred in a colony carrying a targeted mutation of *Mab21l4* (also known as *2310007B03Rik*). The mutation was detected through observation of occasional mice in this colony with raised ABR thresholds to low frequency stimuli that did not segregate with the targeted mutation in *Mab21l4*. The mutation was identified as a point mutation in *Klhl18* by positional cloning (g.9:110455454C>A), predicted to cause a Val55Phe amino acid change in the BTB domain of the protein [2]. The second was a targeted mutant allele (*Klhl18^tm1a(KOMP)Wtsi*, referred to here as *Klhl18^tm1a*, Fig 1B) carrying a promoter-driven knockout-first allele, with a large cassette inserted between exons 5 and 6, which interferes with transcription leading to knockdown of expression [4,15]. The inserted cassette contains a β-galactosidase/LacZ reporter gene. Further details can be found at www.mousephenotype.org. Colonies were maintained by heterozygous intercrosses or homozygotes crossed with heterozygotes. Both *Klhl18* mutant lines produce homozygous offspring that are viable and fertile, and hearing impairment was inherited in a recessive manner. Both males and females were included as no difference in auditory function was noted between them. These mutant mice are available through the European Mouse Mutant Archive (EMMA).

## Genotyping

*Klhl18^lowf* mice were genotyped by PCR amplification followed by restriction enzyme digest of the PCR product. The *Klhl18^lowf* mutation is a C>A missense mutation in exon 10 of the gene [2]. We amplified genomic DNA in this region (forward primer sequence: GCACAATGGTAG GGGTTCAG; reverse primer sequence, GCAGTGTCGCTCAATATTTGTCTTTGTATTCTCTTT GGCCCACAGATTGGGGACCACAAGTTCAGTGCTCACCAG). The PCR primers used together with the point mutation generated a Bgl II restriction site (AGATCT) in the mutant PCR product while the corresponding wildtype allele sequence (AGATCC) was not recognised. BgI II was used to digest the PCR product yielding 2 sequences of 78 bp and 127 bp from the mutant allele and one sequence of 205 bp from the wildtype allele, and these products were identified by gel electrophoresis. In some cases, genotypes were obtained by sequencing the PCR product amplified from a separate pair of primers (forward sequence GAAGTGTAAGGGTGTGGGGG, reverse sequence CTTGCCCGTCTGTCATCCC). The reverse primer was used for sequencing.

*Klhl18*$^{tm1a}$ mice were genotyped using a common forward primer (`CCTGTGACAAGCAG` `TCTGAAGG`), a wildtype reverse primer (`TGCTAGGGAGTGAATCTAGGGC`) and a mutant-specific reverse primer (CasR1 `TCGTGGTATCGTTATGCGCC`). The resulting band sizes were 524 bp for the wildtype product and 384 bp for the mutant product. Primers specific for the Neomycin resistance gene in the introduced cassette were also used to detect the presence or absence of the inserted DNA of the mutant allele (Fig 1) [4,15].

## Anaesthesia

In experiments where mice were tested longitudinally at different ages, mice were anaesthetised by intra-peritoneal injection of 100 mg/kg ketamine (Ketaset, Fort Dodge Animal Health) and 10 mg/kg xylazine (Rompun, Bayer Animal Health) and recovery was promoted using 1 mg/kg atipamezole (Antisedan, Pfizer). For terminal experiments, mice were anaesthetised with intra-peritoneal urethane (0.1 ml/10 g bodyweight of a 20% w/v solution of urethane in water).

## Auditory Brainstem Response (ABR) recordings

Brainstem auditory evoked potentials were measured using the method described in detail previously [16,17]. Anaesthetised mice were placed in a sound-attenuating chamber (IAC Acoustics Limited) and subcutaneous recording needle electrodes (NeuroDart; Unimed Electrode Supplies Ltd, UK) were inserted on the vertex and overlying the left and right bullae. Responses were recorded to free-field calibrated broadband click stimuli (10 μs duration) and tone pips (5 ms duration, 1 ms onset and offset ramp) at frequencies of 6, 12, 18, 24 and 30 kHz, at levels ranging from 0–95 dB SPL (in 5 dB steps) at a rate of 42.6 stimuli per second. Stimuli were generated via custom software on a RZ6 multifunction processor (Tucker Davis Technologies, TDT) and presented via a FF1 loudspeaker (TDT). Evoked responses were amplified, digitized, and bandpass filtered between 300–3000 Hz, using custom software and TDT hardware (RZ6 processor, RA4LI low impedance headstage, RA4PA preamplifier). Thresholds of ABRs were defined as the lowest stimulus level to evoke a visually identifiable waveform (S1 Fig). Evoked ABR waveforms were analysed offline and the latency and amplitude of ABR waves 1–4 were plotted as a function of sound level.

## Distortion Product Otoacoustic Emissions (DPOAE) recordings

We made measurements of the 2f1-f2 DPOAE component in mice across a range of ages, either as a terminal experiment in different cohorts of mice, or as part of longitudinal experiments in the same animals at increasing ages. In all cases, measurements were made inside a sound-attenuating chamber (IAC Ltd) with the mouse positioned on a heating blanket. In terminal experiments, urethane-anaesthetised mice had their left pinna and cartilaginous ear canal removed before a hollow conical speculum was positioned to give an unobstructed view of the tympanic membrane. The DPOAE measurement probe (see later), with a small rubber gasket close to its tip was then positioned within the speculum. For longitudinal experiments, ketamine/xylazine-anaesthetised mice were placed in a prone position and the head was tilted approximately 45˚ such that the left ear was uppermost. The DPOAE probe assembly was positioned vertically, such that the probe tip was sitting just behind the tragus, with the probe pointing down towards the opening of the ear canal. The DPOAE probe assembly used here was comprised of a pair of EC1 electrostatic drivers (TDT) coupled to the guide tubes of an ER10B+ low noise DPOAE system (Etymotic Inc) via 5 cm plastic tubes.

Stimuli were generated and DPOAE responses recorded using a RZ6 multifunction processor (TDT), under the control of TDT BioSigRZ software (TDT). Continuous f1 and f2 tone

stimuli were generated and presented via different EC1 drivers within the DPOAE probe. Frequencies for f2 were set to match the ABR tone-pip frequencies used (i.e. 6, 12, 18, 24 and 30 kHz). The f2 tone was presented at a frequency 1.2 x f1, and a level 10 dB SPL lower than f1. Sound pressure levels of the f2 stimulus ranged from -10 dB to 65 dB in 5 dB steps. ER10B + microphone signals recording during stimulus presentation were digitised at a sampling rate of 195312.5 Hz on the RZ6 processor for online Fast Fourier Transformation (FFT) to yield a power spectrum containing the f1 and f2 stimulus components and the main 2f1-f2 DPOAE component of interest in this study (S2 Fig).

From each FFT trace recorded, a number of parameters were calculated; the 2f1-f2 DPOAE amplitude, the mean noise-floor amplitude (as the average of the 20 spectral lines surrounding the DPOAE frequency) and two-times the standard deviation (SD) of the noise-floor mean. These values are plotted across stimulus level to produce plots from which the threshold of the DPOAE was defined as the lowest stimulus level when the DPOAE amplitude exceeded the 2 SDs above the recording noise-floor. From these data, we calculated mean DPOAE growth functions for each f2 stimulus frequency for control and mutant mice and mean DPOAE thresholds for these frequencies in control and mutant mice.

## Endocochlear Potential (EP) recordings

The positive potential within the scala media of *Klhl18*[lowf] mice was measured in urethane-anaesthetised mice as described previously [18,19]. A reference electrode (Ag-AgCl pellet) was positioned under the skin of the neck. A small hole was made in the basal turn lateral wall and the tip of a 150 mM KCl-filled glass micropipette was inserted into the scala media. The EP was recorded as the differential potential between the tip of the glass electrode and the reference electrode.

## Round Window Response (RWR) recordings

Measurements of evoked potentials detected at the round window of the cochlea were made in *Klhl18*[lowf] mice aged 6 weeks. The methods used here were modified from those of Harvey and Steel [20]. In urethane-anaesthetised mice, following insertion of a tracheal cannula and placement in a custom-built head-holder, a subdermal needle reference electrode was inserted on the midline of the neck. A further subdermal needle electrode was inserted in the skin at the base of the tail to serve as a ground electrode. After removing the pinna, a small opening was made in the bulla and the exposed tip of a Teflon-coated silver wire was positioned on the round window membrane.

Auditory stimuli were generated using custom software and RZ6 multifunction processor (TDT) and presented free-field via a MF1 magnetic loudspeaker (TDT) positioned 15 cm away from and directly opposite the exposed ear canal. Stimuli used were 20 ms duration tone pips, with a 1 ms ramp at the onset and offset, of the same frequencies used in the previous *in vivo* measurements (6, 12, 18, 24 and 30 kHz), with a fixed starting phase of 90°. Tone pips were presented from 0–95 dB SPL, at a repetition rate of 10.65 stimuli/second, with the onset of the tone delayed by 5 ms after the onset of recording. For each sound level, 256 presentations of the tone pip contributed to the generation of an averaged evoked RWR. Signals detected at the round window were amplified (x1000 gain) and filtered (1 Hz high pass, 50 kHz low pass) using a DP311 differential amplifier (Warner Instruments). The amplified filtered signal was sampled at 97656.25 Hz by the RZ6 processor (TDT) and under software control, 80 ms snippets of the signal were used to form an averaged RWR to each combination of tone frequency and sound level.

## Analysis of electrophysiological responses from the round window of *Klhl18* mice

The RWRs recorded were analysed offline (S3 Fig) using custom-written scripts in Matlab (v2019a, The Mathworks Inc., Natick MA, USA). Each RWR (S3A Fig) was filtered into narrow bands to extract different components of the signal. A low pass filter (corner frequency, Fc, 3000 Hz) was used to extract a combined cochlear nerve Compound Action Potential (CAP) and Summating Potential (SP) response (S3Bi Fig). The SP was measured as the mean amplitude of the waveform calculated over the 19–24 ms time window (equivalent to the 14–19 ms section of the tone burst stimulus). The SP could be either positive or negative in sign, relative to the zero baseline of the response (S3Bii Fig). The amplitude of SP measured across stimulus level (from 0–95 dB) was plotted to form an input-output function (S3Biii Fig). A bandpass filter (high pass Fc, 300 Hz; low pass Fc, 3 kHz) was used to extract the cochlear nerve CAP (S3Ci Fig). The CAP was formed of significant negative and positive peaks to its waveform (labelled N and P, respectively in S3Ci and S6-1Cii Figs). For each stimulus level, the N-P peak-to-peak amplitude was calculated and along with the N and P latencies following the onset of the tone pip were plotted to form input-output functions for these parameters (S3Ciii and S3Civ Fig). A narrow bandpass filter centred at stimulus frequency (with low and high pass Fc's of ± 100 Hz) was used to isolate the Cochlear Microphonic (CM) (S3Di Fig). This filtered response was trimmed to a time window equivalent to the 2–18 ms section of the tone burst stimulus, and this steady-state section of the CM response was subject to Fast Fourier Transformation. The resultant power spectrum (S3Dii Fig) was used to determine the amplitude of the CM component. The amplitude of the CM was converted to a dB scale, using 1 μV as a reference amplitude. The dB (re 1 μV) CM amplitude measured across stimulus level was plotted to form an input-output function of CM growth (S3Diii Fig).

## Innervation and synaptic labelling with confocal microscopy imaging

Inner ears from 6-week old mice were fixed in 4% paraformaldehyde for 2 hours and decalcified in 0.1 M EDTA overnight at room temperature. Following fine dissection in PBS, the organ of Corti was permeabilised in 5% Tween20 in PBS for 40–60 minutes and incubated in blocking solution (0.45% Triton X-100, 10% normal horse serum in PBS) for 2 hours. Following blocking, the samples were incubated overnight at room temperature with shaking using primary antibodies in 0.36% Triton, 6% normal horse serum in PBS. For synaptic labelling, primary antibodies were mouse anti-GluR2 (1:200, MAB397, Emd Millipore) and rabbit anti-Ribeye (1:500, 192 103, Synaptic Systems). For neuron labelling, primary antibodies were mouse anti-CtBP2 (1:400, BD Transduction Laboratories 612044) to label pre-synaptic ribbons and IHC nuclei and chicken anti-Neurofilament-Heavy (1:800, Abcam ab4680) to label unmyelinated neural dendrites. Samples were then washed with PBS and then incubated with secondary antibodies for 45–60 minutes in the dark. For synapses, secondary antibodies were either donkey anti-mouse IgG Alexa Fluor594 (A21203, ThermoFischer Scientific) or goat anti-mouse IgG2a Alexa Fluor488 (A21131, ThermoFischer Scientific) and goat anti-rabbit IgG Alexa Fluor546 (A11035, ThermoFischer Scientific). For neuron labelling, secondary antibodies were donkey anti-mouse Alexa Fluor594 (1:500, A-21203, Molecular Probes) and goat anti-chicken Alexa Fluor488 (1:300, A11039, Life technologies). Samples were then washed in PBS before mounting in ProLong Gold mounting medium with DAPI and stored at 4˚C in the dark. Specimens were viewed using a Zeiss Imager 710 confocal microscope interfaced with ZEN 2010 software with a x63 objective and numerical aperture of 1.4. The whole length of the organ of Corti was examined and images collected at best-frequency regions based upon the frequency-place map of Müller and colleagues [21]. Z-stacks were taken at 0.25 μm intervals

and maximum intensity projection images were generated. Brightness and contrast were normalised for the dynamic range in all images. The number of ribbon synapses per IHC was quantified by manually counting the co-localised Ribeye and GluR2 puncta in the confocal maximum projection images and dividing the sum by the number of DAPI-labelled IHC nuclei. An average of six IHCs per image was considered for ribbon synapses counting using the cell-counter plugin in Fiji software.

## Scanning electron microscopy

Cochlear samples from *Klhl18*$^{lowf}$ mice aged 3 weeks and 4 weeks old were fixed in 2.5% glutaraldehyde in 0.1 M sodium cacodylate buffer with 3 mM calcium chloride, dissected to expose the organ of Corti, then processed by a standard osmium tetroxide-thiocarbohydrazide (OTOTO) protocol [22]. After dehydration, samples were subjected to critical point drying, mounted and viewed using a Jeol JSM-7800F Prime Schottky field emission scanning electron microscope. An overview of the cochlea was imaged to allow calculation of percentage distances along the cochlear duct to superimpose the frequency-place map [21], allowing subsequent imaging of consistent locations across different specimens. Images were assessed by three independent viewers who were blinded to genotype.

## Experimental design and statistical analysis

We used heterozygotes as controls in most of the electrophysiological experiments because we did not see any obvious difference between heterozygotes and wildtype littermates. There was no significant effect of sex on any of the physiological recordings in control or mutant mice of either *Klhl18*$^{lowf}$ or *Klhl18*$^{tm1a}$ lines (S4A–S4F Fig; S1 Table), so both males and females were combined into a single group for their age and genotype. As the auditory phenotype changes with age of the homozygous mutants, we use littermates as controls and carefully age-matched mice in different age groups; 2 weeks (Post-natal day, P, 14 +/- 0 days); 3 weeks (P21 +/- 0 days); 4 weeks (P28 +/- 1 day); 6 weeks (P42 +/- 2 days); 8 weeks (P56 +/- 4 days); 14 weeks (P98 +/- 4 days).

All statistical comparisons were made using the analysis routines within Graphpad Prism (v8.4.2). Auditory Brainstem Response, Distortion Product Otoacoustic Emissions and Round Window Response data were analysed using a Mixed-effect Model, with either Sidak's or Tukey's multiple comparisons tests. Endocochlear Potential values were compared using a Kruskal-Wallis One-Way ANOVA. Counts of synaptic components were compared using a two-tailed unpaired t-test, with Welch's correction. All tests were performed with alpha = 0.05. Statistical significance was determined at the level of $p < 0.05$.

## Results

### *Klhl18* mutant mice showed progressive increases in ABR thresholds with age, particularly for low frequency stimuli

The ABR is a convenient electrophysiological approach that can be recorded non-invasively, allowing us to track any changes in auditory responses in the same mouse as it ages. In heterozygous control mice, ABR thresholds improved from 2 weeks old, through 3 weeks old to 4 weeks old (Fig 2A–2C), indicating maturation of auditory sensitivity. At 2 weeks old, ABR thresholds of *Klhl18*$^{lowf/lowf}$ mutants were slightly but statistically significantly elevated, compared to age-matched heterozygous littermate controls (Fig 2A; S1 Table). At 3 weeks old, ABR thresholds of mutant and control mice are generally comparable (Fig 2B). From the age of 4 weeks (Fig 2C–2F), ABR thresholds for all stimuli were significantly higher in mutants than in age-matched

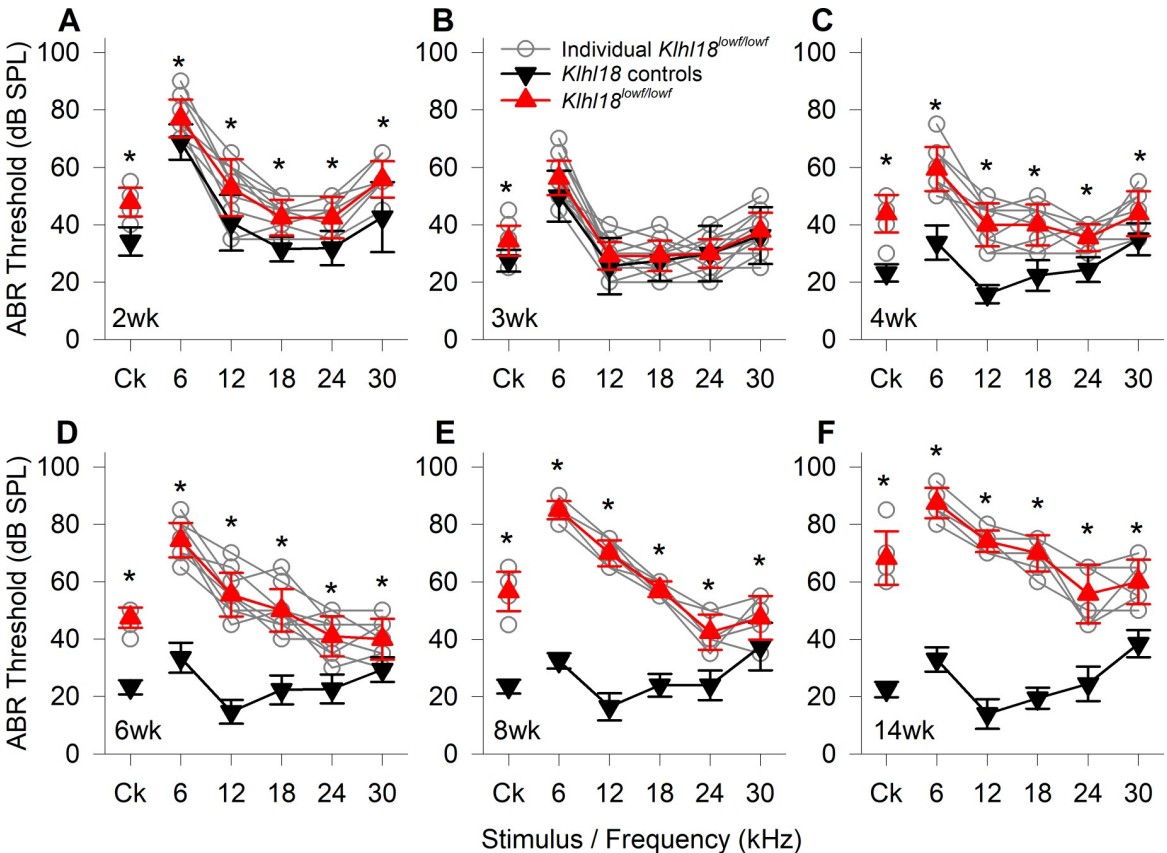

**Fig 2. Age-related changes in ABR thresholds of *Klhl18^lowf* mice.** A summary of ABR thresholds estimated from control (all heterozygotes) and mutant (homozygotes) are shown for *Klhl18^lowf* mice aged 2 weeks (**A**, n = 13 control, n = 12 mutant), 3 weeks (**B**, n = 8 control, n = 19 mutant), 4 weeks (**C**, n = 17 control, n = 9 mutant), 6 weeks (**D**, n = 17 control, n = 10 mutant), 8 weeks (**E**, n = 10 control, n = 6 mutant) and 14 weeks (**F**, n = 10 control, n = 6 mutant). Thresholds from individual mutant mice are indicated by grey circles. Mean threshold (±SD) are plotted for control mice as black down-triangles and for mutant mice as red up-triangles. Ck: Click stimulus. The results of a mixed-effects model statistical analysis between control and mutant thresholds, followed by Sidak's multiple comparisons test was used to examine the difference between thresholds for each stimulus are shown in S1 Table. Significant differences are indicated here by *.

heterozygous controls (See S1 Table). As the mice aged beyond 4 weeks, control mice maintain low ABR thresholds but there was a progressive elevation of thresholds from 4 weeks old to 14 weeks in the *Klhl18^lowf* mutant mice, most pronounced at 6–18 kHz (Fig 2C–2F). A similar pattern of age-related threshold changes is seen in *Klhl18^tm1a* mutant mice (S5 Fig).

### *Klhl18* mutant mice showed abnormal ABR waveform shapes

In addition to determining the threshold for detection of an ABR, we also examined the waveform features. Group averaged ABR waveforms from *Klhl18^lowf* mice for selected age groups and frequency stimuli are plotted in Fig 3. As the ABR waveform changes with increasing stimulus level (amplitude increases and latency decreases and the stimulus increases above threshold), we plot mean waveforms at both a fixed dB SPL (65dB SPL; Fig 3A–3I) and also at 20 dB sensation level (dB SL; dB above threshold; Fig 3J–3R). For mice aged 2 weeks old, waveforms evoked by click stimuli and 12 kHz and 24 kHz tone pips (Fig 3A–3C and 3J–3L) showed that *Klhl18* mutant mice had smaller amplitudes than controls. At 3 weeks old, when ABR thresholds were generally comparable across genotypes, the averaged waveforms for these stimuli

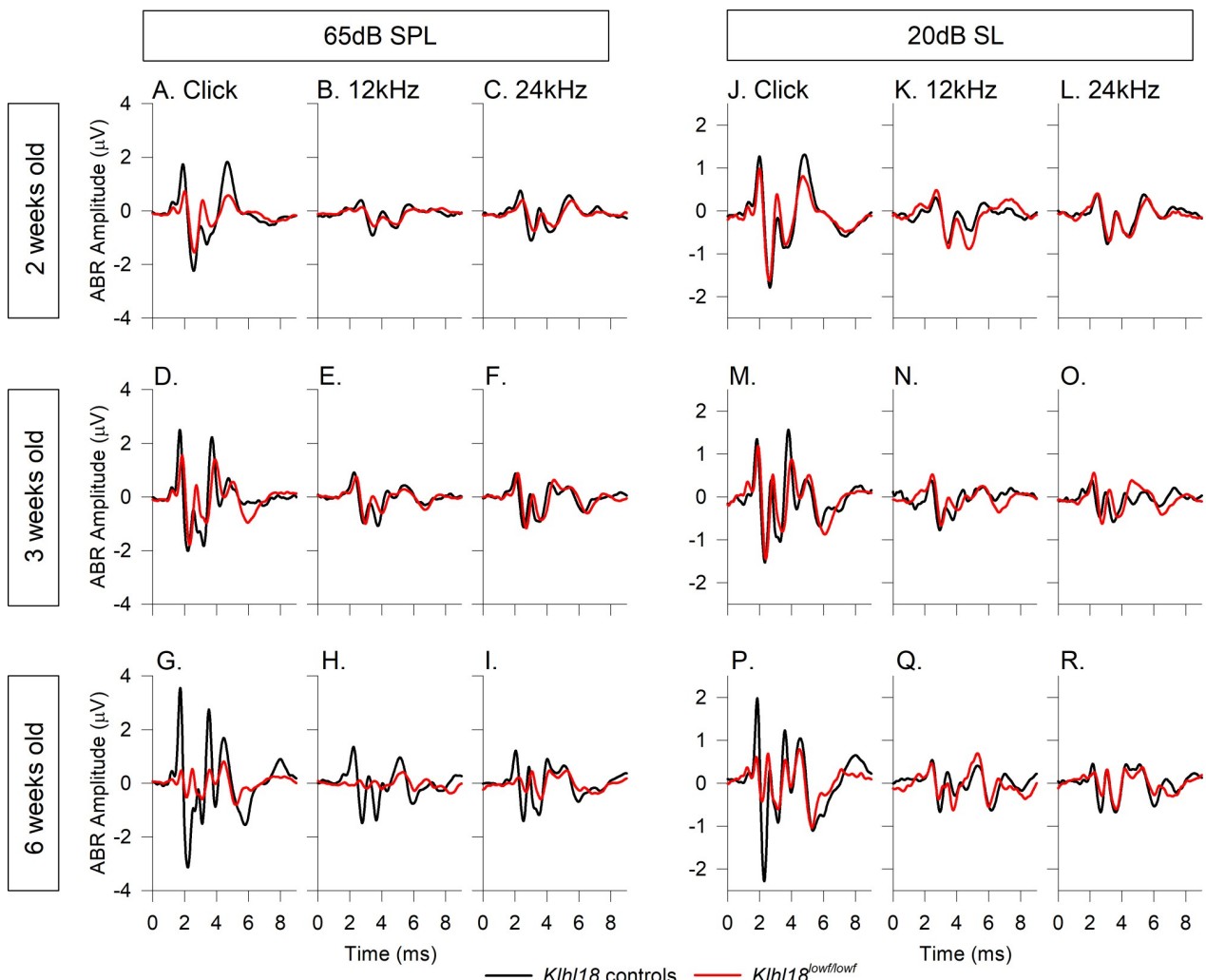

**Fig 3. Mean ABR waveforms of *Klhl18^lowf* mice at 65dB SPL and 20 dB SL.** Group averaged ABR waveforms, evoked by 65 dB SPL stimuli, are plotted for mutant mice (red lines) and heterozygous littermate controls (black lines) aged 2 weeks (**A-C**), 3 weeks (**D-F**) and 6 weeks (**G-I**) in response to click stimuli (**A,D,G**), 12 kHz stimuli (**B,E,H**) and 24 kHz stimuli (**C,F,I**). Group averaged ABR waveforms, evoked by stimuli presented at 20 dB above threshold (20 dB sensation level, dB SL), are plotted for mice aged 2 weeks (**J-L**), 3 weeks (**M-O**), 6 weeks (**P-R**) in response to click stimuli (**J,M,P**), 12 kHz stimuli (**K,N,Q**) and 24 kHz stimuli (**L,O,R**). Group sizes were, at 2 weeks old, n = 13 control mice, n = 12 mutants; at 3 weeks old, n = 8 control mice, n = 19 mutants; at 6 weeks old, n = 17 controls, n = 10 mutants.

were more similar (Fig 3D–3F and 3M-3O); the click-evoked response was smaller in mutants, but the tone-evoked responses had similar amplitudes in control compared with mutant mice. These mean waveform amplitudes from 2–3 weeks old mice were in contrast to those from older mice. At 6 weeks old, when mutant mice exhibited a clear increase in thresholds, the averaged waveforms of mutants evoked by click, 12 kHz and 24 kHz stimuli were much smaller than those in control mice (Fig 3G–3I and 3P–3R). Similar observations were obtained from *Klhl18^tm1a* mutant mice (S6 Fig).

## ABR wave 1 showed reduced amplitude and longer latency in *Klhl18* mutant mice

In order to quantify the abnormal features of the ABR waveform, ABR wave 1 amplitudes (measured as wave P1 –N1 peak to peak amplitude) and P1 latencies were plotted as a function

of stimulus level (Fig 4; See also S1 Fig). The mild threshold elevation in 2-week old *Klhl18*$^{lowf}$ mutants was reflected in a shift to the right of growth curves (Fig 4A–4C and 4J–4L). In mutant mice aged 3 weeks old, wave 1 amplitude was similar to controls (Fig 4D–4F). However, P1 latency maintained a slight increase in mutants compared to age-matched heterozygous controls (Fig 4M–4O). By 6 weeks old, when ABR thresholds were clearly raised in mutants, much larger changes in ABR wave 1 amplitude and latency were noted (Fig 4G–4I and 4P–4R). Similar patterns of amplitude and latency changes were observed in *Klhl18*$^{tm1a}$ mutant mice (S7 Fig). Amplitudes and latencies of ABR wave 1 in *Klhl18*$^{tm1a}$ heterozygotes were similar to their wildtype littermates (S8 Fig).

## Normal DPOAE responses in *Klhl18* mutant mice

Auditory Brainstem Responses reflect auditory nerve and brainstem responses, but DPOAEs are useful additional measures because they reflect outer hair cell function [23]. At 2 weeks old, there was a small statistically significant reduction in DPOAE amplitudes and elevation in DPOAE thresholds (for f2 frequencies of 12–30 kHz) in *Klhl18*$^{lowf}$ mutants, similar to the small differences in ABR responses recorded at this age (Fig 5A–5F). In mice aged 3 weeks old, DPOAE responses were the same in *Klhl18*$^{lowf}$ mutants as in heterozygous littermate controls, again mimicking the ABR thresholds at 3 weeks old (Fig 5G–5L; no significant interaction of either DPOAE amplitude or threshold with genotype; see S1 Table). However, in contrast to the significant elevation in ABR thresholds by the age of 6 weeks (Fig 3), there was no difference between DPOAE responses of control and mutant mice at this age (Fig 5M–5R; no significant interaction of either DPOAE amplitude or threshold with genotype; S1 Table). Furthermore, no differences were found in DPOAEs in *Klhl18*$^{tm1a}$ mutant mice up to 6 weeks old (S9 Fig).

## Endocochlear Potential (EP) was normal in *Klhl18*$^{lowf}$ mutant mice

Some mutations lead to reduced endocochlear potential which can cause hearing impairment [18], so we measured EP in 6 weeks old *Klhl18*$^{lowf}$ mutant and littermate control mice. EP was 122.4 ± 5.4 mV (mean ± SD) in 6 wildtype mice (aged 42.0 ± 0 days), 121.2 ± 5.3 mV in 6 heterozygote mice (aged 42.3 ± 1.2 days) and 123.4 ± 5.6 mV in 7 homozygous mutant mice (aged 42.1 ± 0.9 days). There was no significant difference in EP between mice of these 3 genotypes (S1 Table).

## Round Window Responses (RWRs) match ABR and DPOAE responses

Additional information about cochlear function can be obtained by recording responses using an electrode placed on the round window of the cochlea, which is closer to the source of the electrical activity. We analysed three different responses extracted from recordings from the round window in 6-week old *Klhl18*$^{lowf}$ mutants and heterozygous littermate controls: Compound Action Potential (CAP), Cochlear Microphonic (CM) and Summating Potential (SP) (Fig 6). Extraction of the different potentials is illustrated in S3 Fig. Examples of RWRs and their components are shown in S10 and S11 Figs for a *Klhl18*$^{+/+}$ wildtype mouse and a *Klhl18*$^{lowf/lowf}$ mutant mouse, respectively, for 18kHz tone stimuli.

Compound action potentials (CAP), reflecting auditory nerve activity [e.g. 24], had thresholds and amplitudes very similar to those seen for ABRs, with low frequencies more severely affected than high frequencies in mutants (compare Fig 6K with Fig 2D, and Fig 6G and 6I with Fig 4H and 4I). Latencies of CAP responses were longer in mutants than in controls (Fig 6A–6E) but the shift in latency at each frequency matched the shift in threshold so the latencies appear to be normal when adjusted for hearing level.

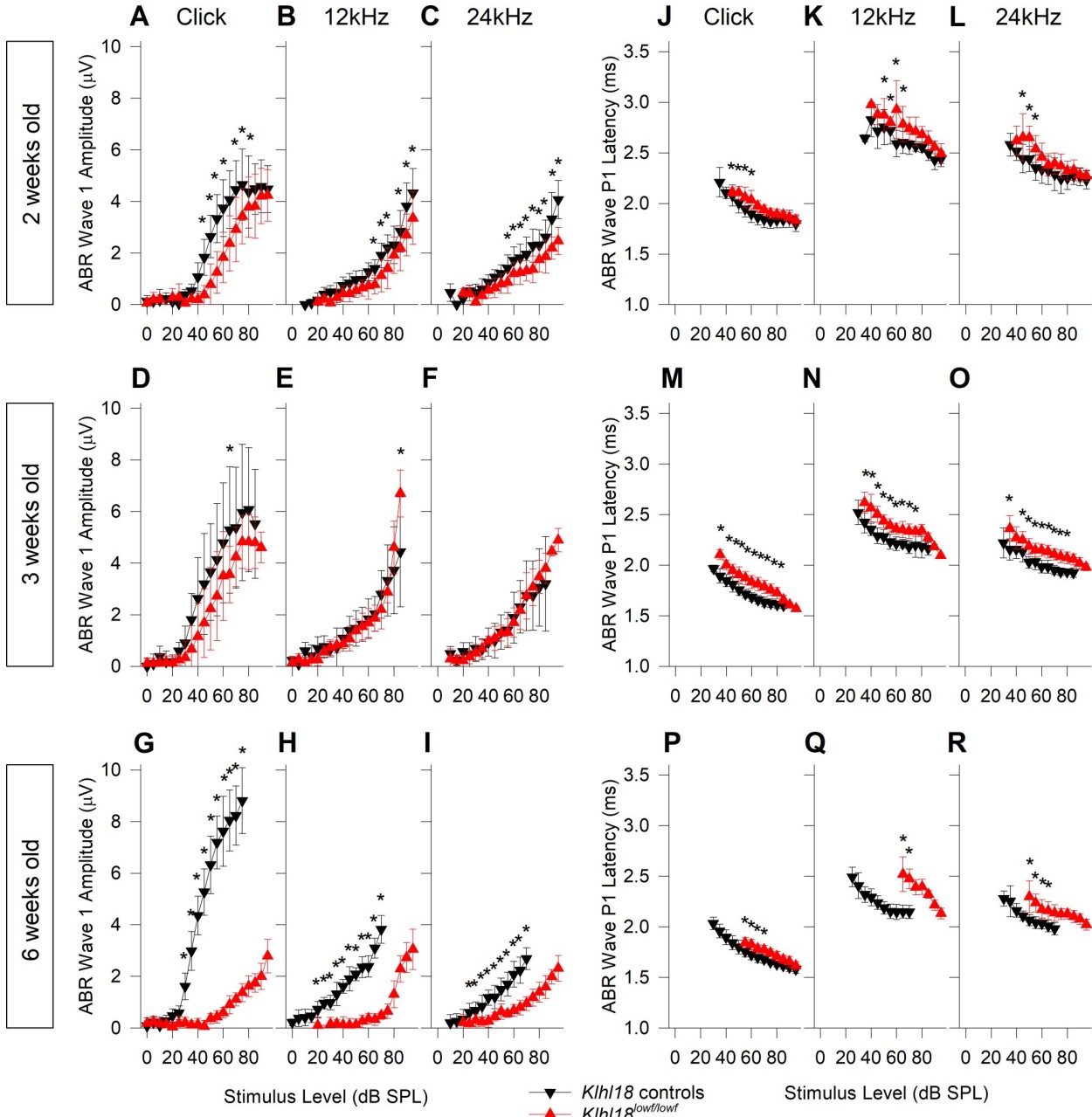

**Fig 4. ABR wave 1 amplitude and latency input-output functions in *Klhl18*<sup>lowf</sup> mice.** Mean ABR wave 1 P1-N1 peak-to-peak amplitudes (±SD) as a function of stimulus level for mutant mice and heterozygous littermate controls aged 2, 3 and 6 weeks old are plotted in **A-C** (n = 13 controls, n = 12 mutants), **D-F** (n = 8 controls, n = 19 mutants) and **G-I** (n = 17 controls, n = 10 mutants), respectively. Results are plotted for click stimuli (**A,D,G**), 12 kHz tones (**B,E,H**) and 24 kHz tones (**C,F,I**). Data from control mice are plotted as black down-triangles. Data from mutant mice are plotted as red up-triangles. Mean ABR wave P1 latency (±SD) as a function of stimulus level is plotted for the same mice aged 2, 3 and 6 weeks old are plotted in **J-L**, **M-O** and **P-R**, respectively. Results are plotted for click stimuli (**J,M,P**), 12 kHz tones (**K,N,Q**) and 24 kHz tones (**L,O,R**). The results of a mixed-effects model statistical analysis between control and mutant thresholds are shown in S1 Table. Sidak's multiple comparisons test was used to examine the difference between control and mutant data for each stimulus. Significant differences are indicated here by *.

Cochlear microphonics (CM), reflecting mostly outer hair cell activity [e.g. 25], showed no significant difference in thresholds or amplitudes in *Klhl18*<sup>lowf/lowf</sup> compared with *Klhl18*<sup>+/+</sup> mice (Fig 6L–6Q, S1 Table). This matched our findings from DPOAEs, which were also normal in the mutants.

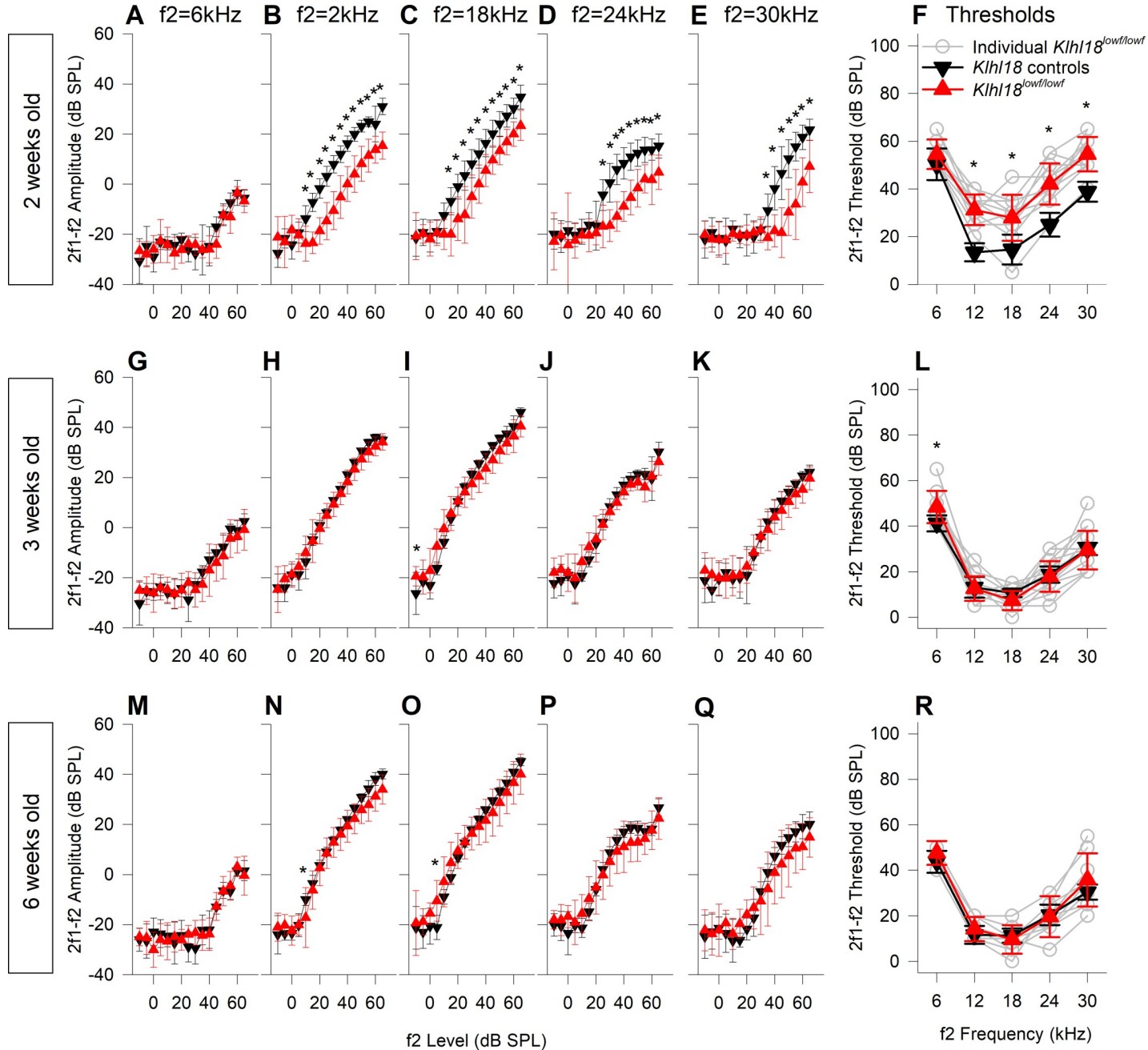

**Fig 5. DPOAE growth and thresholds in *Klhl18^lowf* mice.** DPOAE results from mutant mice and heterozygous littermate controls aged 2, 3 and 6 weeks old are plotted in **A-F** (n = 13 controls, n = 12 mutants), **G-L** (n = 8 controls, n = 19 mutants) and **M-N** (n = 9 controls, n = 13 mutants), respectively. Data from control mice are plotted as black down-triangles. Data from mutant mice are plotted as red up-triangles. The mean (±SD) amplitude of the 2f1-f2 DPOAE (dB SPL) is plotted as a function of f2 stimulus level (dB SPL) for f2 frequencies of 6 kHz (**A,G,M**), 12 kHz (**B,H,N**), 18 kHz (**C,I,O**), 24 kHz (**D,J,P**) and 30 kHz (**E,K,Q**). Mean threshold (±SD) of the 2f1-f2 DPOAE (derived from individual growth functions, e.g. shown in S2 Fig) and plotted in **F**, **L** and **R** for mice aged 2 weeks, 3 weeks and 6 weeks respectively. In addition to the mean data, thresholds from individual mutant mice are plotted as grey open circles. Results from a mixed-effects model statistical analysis between control and mutant data are shown in S1 Table. Sidak's multiple comparisons test was used to examine the difference between control and mutant data for each stimulus. Significant differences are indicated here by *.

Summating potentials (SP) are a dc shift in the round window recording sustained for the duration of a toneburst and are thought to reflect hair cell depolarisation, particularly of basal turn IHCs [25,26]. They can be negative or positive in polarity, with negative SP usually

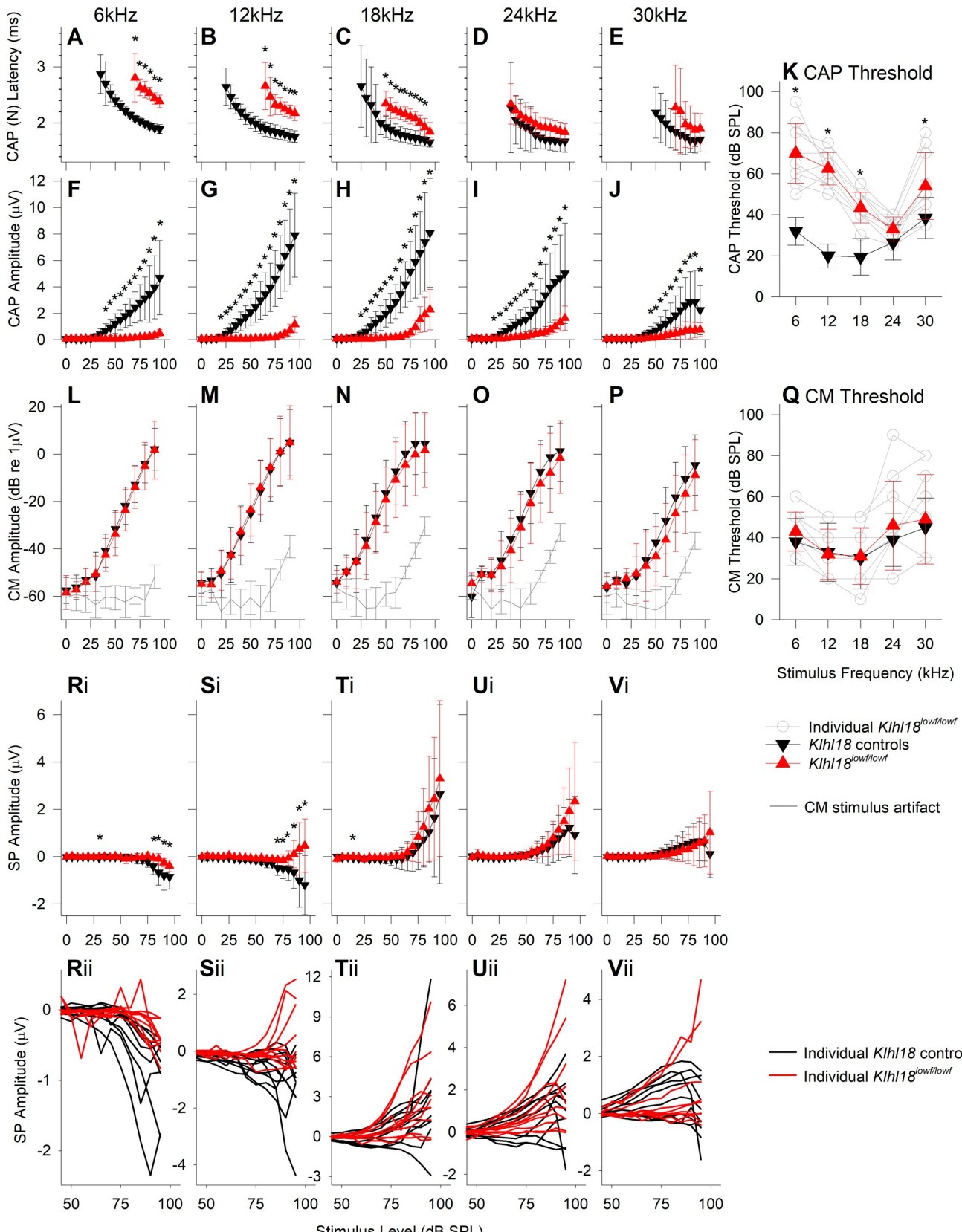

**Fig 6. Round window response measurements from 6-weeks old *Klhl18*$^{lowf}$ mice.** RWRs were obtained from 10 wildtype (*Klhl18*$^{+/+}$, aged 41.8 ± 0.6 days), 7 heterozygote (*Klhl18*$^{+/lowf}$ aged 42.6 ± 1.1 days) and 10 homozygote mutant (*Klhl18*$^{lowf/lowf}$ aged 41.3 ± 1.2 days) mice. CAP latency (of wave N) and amplitude (N-P amplitude) are plotted in **A-E** and **F-J**, respectively, for potentials measured in response to tones of 6, 12, 18, 24 and 30 kHz. Data are plotted as mean ± SD for wildtype control mice (black down-triangles) and homozygote mutant mice (red up-triangles). **K**. Mean ± SD of the CAP threshold is plotted against stimulus frequency for wildtype control mice (black down-triangles) and homozygote mutant mice (red up-triangles). Open circles and grey lines indicate CAP thresholds for individual mutant mice. **L-P** plot mean (± SD) CM amplitude (dB re 1 μV) against stimulus level (dB SPL) for wildtype mice (black down-triangles) and homozygote mutant mice (red up-triangles). We were unable to completely remove stimulus artifact from the RWRs recorded in our system. To account for this, we assessed the magnitude of the stimulus artifact across frequency and sound level, from analysis of measurements taken after the mouse was removed the system and the recording electrodes placed in 150 mM KCl. The biological CM was orders of magnitude greater in amplitude than the stimulus artifact and thus it was distinguishable from the artifact. The artifact estimated across all experiments, across stimulus frequency and level are plotted as grey lines (with SD error bars). **Q**. Mean ± SD of the estimated CM threshold is plotted against stimulus frequency for wildtype control mice (black down-triangles) and homozygote mutant mice (red up-triangles). Open circles and grey lines indicate CM thresholds for individual mutant mice. **Ri-Vi** plot mean (± SD) Summating Potential amplitude (μV) against stimulus level (dB SPL) for wildtype mice (black down-triangles) and homozygote mutant mice (red up-triangles). **Rii-Vii** plot SP input-output functions (IOFs) for individual control (black lines) and mutant (red lines) mice. Data from *Klhl18*$^{+/lowf}$ mice are shown in S12 Fig. Results of a mixed-effects model statistical analysis between wildtype, heterozygote (S12 Fig) and homozygote data are shown in S1 Table. Sidak's multiple comparisons test was used to examine the difference between control and mutant data for each stimulus. Significant differences are indicated here by $^*$.

detected at lower frequencies and higher stimulus intensities. We found these responses to be variable between individual mice, whatever their genotype (Fig 6Rii–6Vii). Despite the variability, SP amplitudes were significantly different in mutants at 6 and 12 kHz, while there was no difference in SP at higher frequencies (See Fig 6 and S1 Table). No significant differences between wildtypes (*Klhl18*$^{+/+}$) and heterozygotes (*Klhl18*$^{+/lowf}$) were found in any of the round window measures (Figs 4–6 and S1 Table).

## Normal innervation of the cochlea

The presence of abnormal ABRs and CAPs together with normal DPOAE and CM responses and normal EPs suggests that *Klhl18* mutant mice exhibit an IHC defect or neural hearing loss, so we looked in more detail at the neural elements of the organ of Corti. We used neurofilament immunolabelling in whole mount organ of Corti preparations to examine the dendrites of cochlear neurons in 6-week old *Klhl18*$^{lowf}$ mutants and heterozygous littermate controls in regions corresponding to 6, 18 and 30 kHz characteristic frequencies (Fig 7). Confocal imaging showed a normal pattern of innervation in all regions examined. To examine synaptic organization in *Klhl18*$^{lowf}$ mice, we immunolabelled GluR2 (a post-synaptic glutamate receptor subunit) and Ribeye (a component of presynaptic ribbon synapses) in the organ of Corti of 6 week old mice (Fig 8). We found no significant differences (see S1 Table) in numbers of ribbons or postsynaptic densities or co-localised labelled puncta (indicating synapses) in these mutants compared with heterozygous littermate controls, either in regions exhibiting raised ABR thresholds (12 kHz region) or relatively normal ABR thresholds (24 kHz region).

## Tapering of tips of stereocilia in *Klhl18*$^{lowf}$ mutants

The stereocilia bundle at the top of each hair cell houses the mechano-electrical transduction apparatus and so is essential for normal auditory function. Hearing impairment often results from abnormal or damaged stereocilia, so we examined the apical surface of the organ of Corti by scanning electron microscopy to look for any signs of hair cell degeneration or other defects. Homozygotes, heterozygotes and wildtypes were studied at two ages (3 weeks and 4 weeks old), corresponding to stages when ABR thresholds were normal (3 weeks old) and when thresholds had started to increase in homozygotes (4 weeks old). In general, the organization of the sensory epithelium, the general patterning of inner and outer hair cells within the epithelium, and the appearance of all the associated supporting cell types were normal in

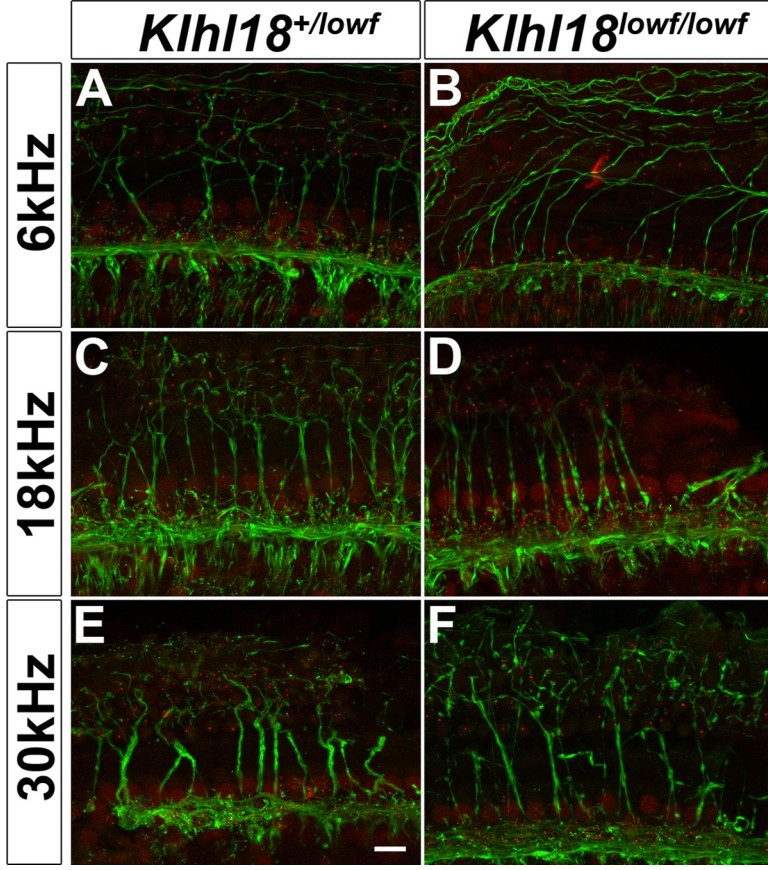

**Fig 7. Auditory nerve innervation of the organ of Corti at 6 weeks old.** Representative confocal microscopy images of the organ of Corti at the 6 kHz (**A,B**), 18 kHz (**C,D**) and 30 kHz (**E,F**) regions of the cochlea from a sample of 7 ears from 5 *Klhl18*<sup>+/lowf</sup> mice (**A,C,E**), and 8 ears from 6 *Klhl18*<sup>lowf/lowf</sup> mice (**B,D,F**). CtBP2-positive regions (IHC nuclei and ribbon synapses) are immuno-stained in red; Neurofilament-positive regions (nerve fibres) are immuno-stained in green. The scale bar shown in (**E**) represents 10μm.

*Klhl18*<sup>lowf</sup> mutant mice. There was no evidence seen of hair cell degeneration in the mutants at these ages, although we did not analyse the lower basal turn.

However, we did observe stereocilia abnormalities in the mutants that were most pronounced in the apical turn. Surprisingly we found similar defects at both 3 and 4 weeks old. The most severe defects were found in IHCs which showed frequent tapering of the tallest stereocilia towards their tips (examples shown in Fig 9A–9H). This feature was variable between hair bundles and also between individual stereocilia within a single bundle (see for example Fig 9D, stereocilia on right side of image). The shorter two rows of stereocilia had a more normal appearance. Outer hair cell stereocilia of mutants looked close to normal, but some bundles appeared slightly more irregular than in wildtypes and there were more often missing individual stereocilia in the shortest row in the mutants (Fig 10A–10H). In both inner and outer hair cells, stereocilia were arrayed in the cuticular plate in the normal V-shaped arrangement for OHCs or crescent for IHCs, and the angle of the V varied along the cochlear duct in a normal way in mutants, with a wider angle of the V in OHCs towards the base. Stereocilia were also arranged in rows of graded heights as usual, but with more irregularity in heights within each row in mutant bundles. All of these hair cell defects in IHCs and OHCs showed a gradient along the length of the cochlear duct with more severe defects towards the

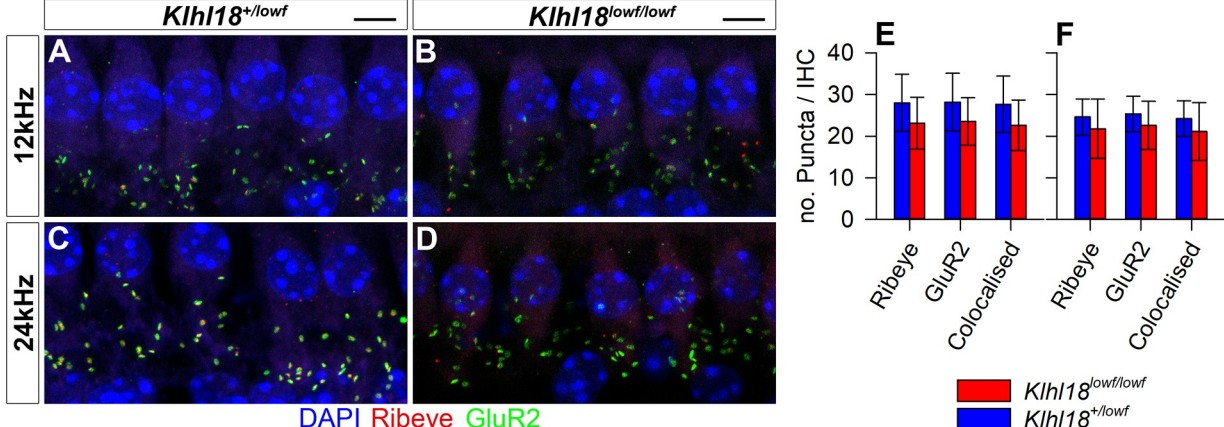

**Fig 8. IHC synapses in *Klhl18* mutant mice at 6 weeks old.** Representative confocal microscopy images of IHCs at the 12 kHz (**A,B**) and 24 kHz (**C,D**) regions of the cochlea from a sample of 7 *Klhl18^{+/lowf}* mice (**A,C**), and 7 *Klhl18^{lowf/lowf}* mice (**B,D**). A row of IHC nuclei can be seen across the upper portion of panels A-D, stained blue with DAPI. Below this row, clusters of immuno-stained puncta are visible. Green puncta show immuno-positive staining for the post-synaptic receptor subunit GluR2. Red puncta show immuno-positive staining for Ribeye, a protein component of the ribbon synapse apparatus found in inner hair cells. Yellow staining represents overlap between GluR2 and Ribeye (colocalised) staining and is presumed to represent a functional synapse between the inner hair cells and the auditory nerve. Scale bars indicate 5 μm. (**E,F**) Counts of Ribeye-positive, GluR2-positive and colocalised Ribeye and GluR2-positive puncta are plotted for the 12 kHz region (E) and the 24 kHz region (F). Data are plotted as bars representing the mean count per IHC (error bars indicate the SD) for control (*Klhl18^{+/lowf}*, blue bars) mice and homozygous mutant mice (*Klhl18^{lowf/lowf}*, red bars).

apex of the cochlea, but the abnormal IHC hair bundles extended further towards the base than abnormal OHC bundles. This tendency to more severe defects in the apex fits with the predominance of ABR threshold increases at low frequencies.

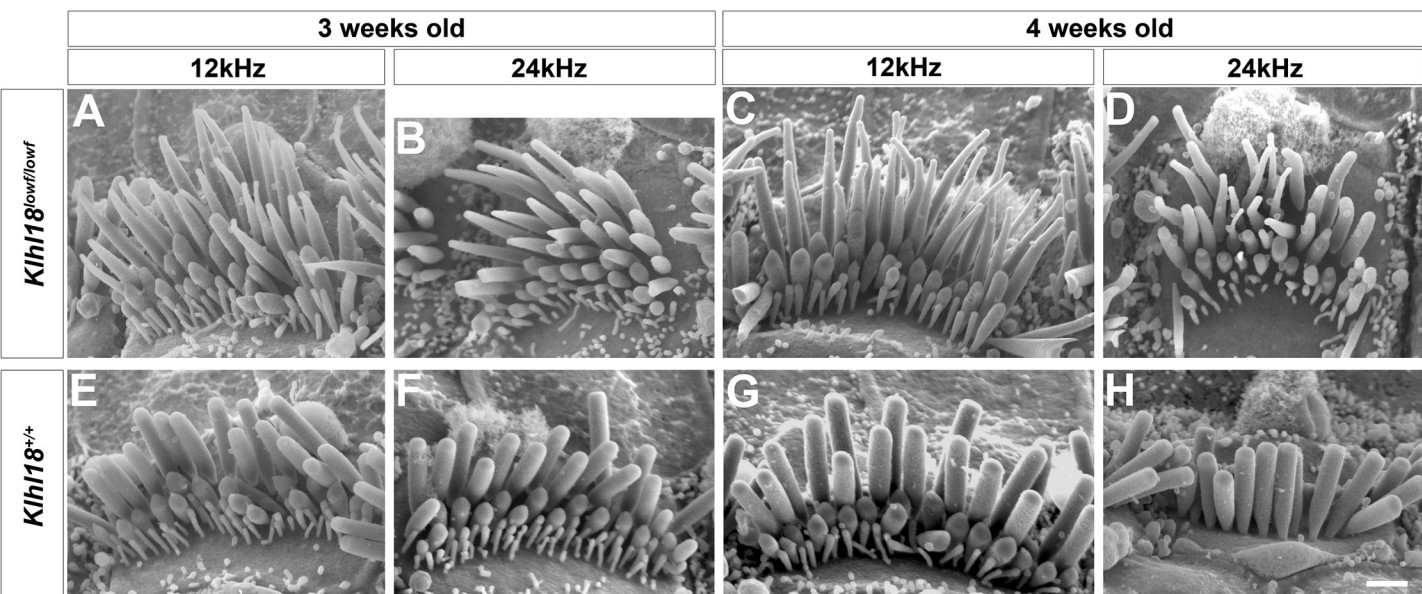

**Fig 9. Scanning electron microscopy of typical stereocilia bundles of inner hair cells in *Klhl18* mutant mice in middle and apical turns.** Representative images are shown from (**A-D**) *Klhl18^{lowf/lowf}* homozygotes and (**E-H**) littermate wildtype mice. (**A, E**) 3 weeks old mice at the 12 kHz best-frequency region; (**B, F**) 3 weeks old mice at the 24 kHz best-frequency region; (**C, G**) 4 weeks old mice at the 12 kHz best-frequency region; (**D, H**) 4 weeks old mice at the 24 kHz best-frequency region. (**A-G**) show views from the modiolar side of the bundle, but (**H**) shows a view from the pillar side to illustrate the normal shape of the tallest row of stereocilia. Scale bar (on panel **H**) represents 1 μm.

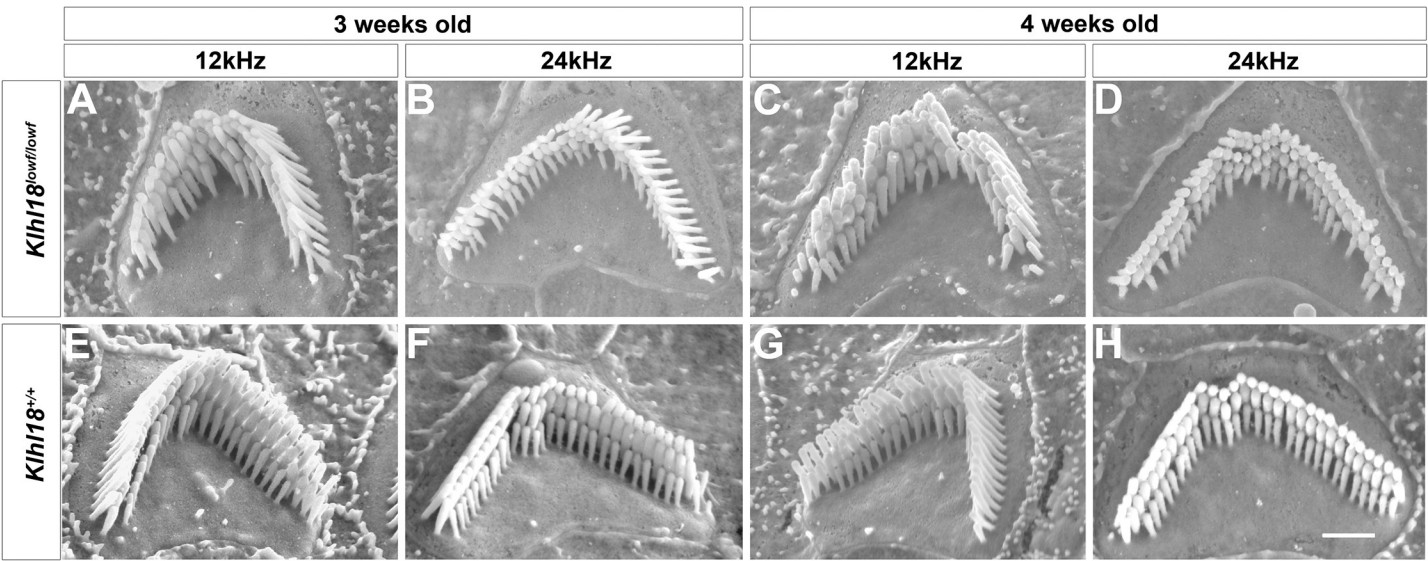

**Fig 10. Scanning electron microscopy of typical stereocilia bundles of outer hair cells in *Klhl18* mutant mice in mid- and apical turns.** Representative images are shown from (**A-D**) *Klhl18^{lowf/lowf}* homozygotes and (**E-H**) littermate wildtype mice. (**A, E**) 3 weeks old mice at the 12 kHz best-frequency region; (**B, F**) 3 weeks old mice at the 24 kHz best-frequency region; (**C, G**) 4 weeks old mice at the 12 kHz best-frequency region; (**D, H**) 4 weeks old mice at the 24 kHz best-frequency region. Scale bar (on panel **H**) represents 1 μm.

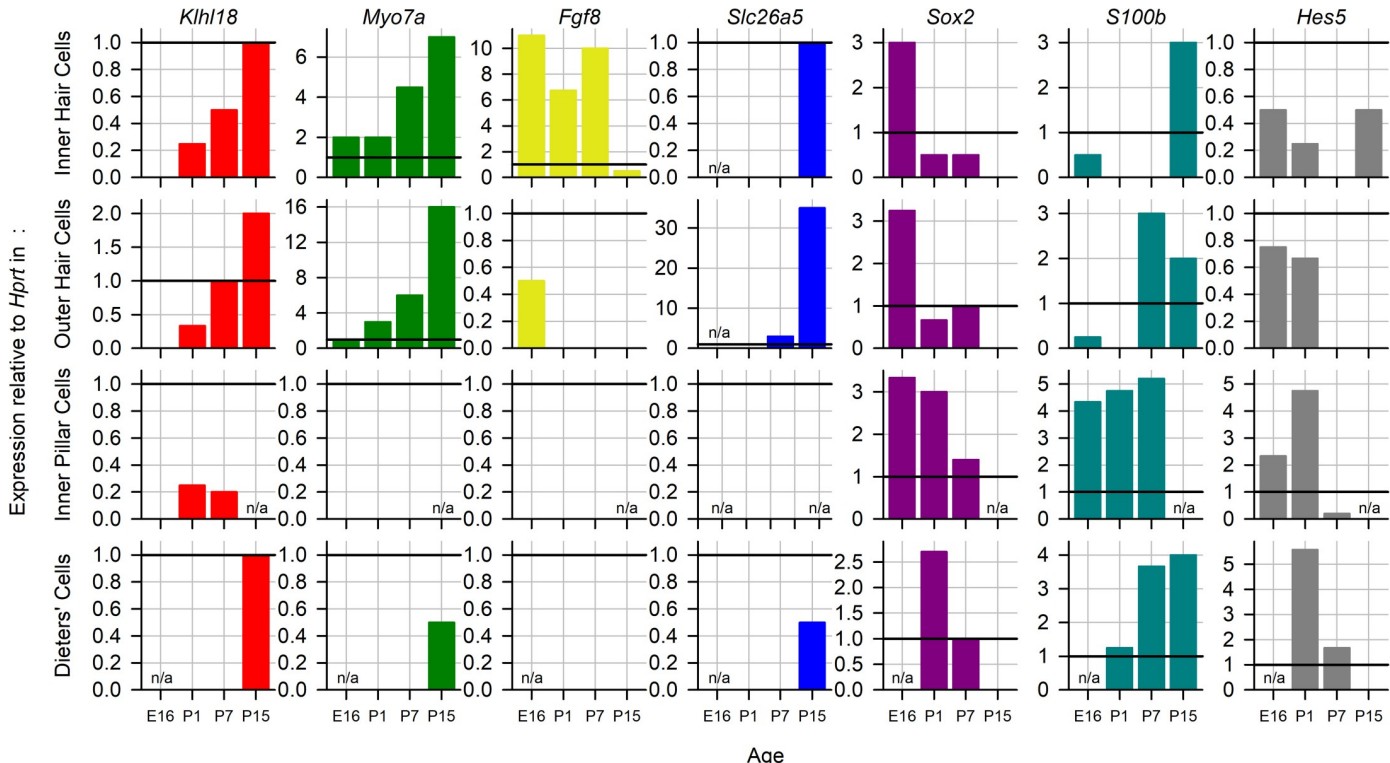

**Fig 11. Expression of *Klhl18* in key cell types during development.** Expression of *Klhl18* (red), *Myo7a* (green), *Fgf8* (yellow), *Slc26a5* (blue), *Sox2* (purple), *S100b* (teal) and *Hes5* (grey) at stages from E16 to P15 in the mouse inner ear. Single cell RNAseq data from [27,28]. Expression levels were normalised to *Hprt* expression, indicated by a horizontal black line at y = 1 on each plot. n/a indicates data not available for this age.

Interestingly, we observed some minor tapering of the distal ends of IHC stereocilia in the extreme apical 10% of the cochlear duct even in wildtypes. Heterozygotes showed an intermediate degree of IHC stereocilia tapering especially visible at 4 weeks old at the 3 and 6kHz best-frequency regions.

## Expression of *Klhl18* in the organ of Corti

To investigate the expression of *Klhl18* in the inner ear, we used single cell RNAseq data from the mouse inner ear at embryonic day(E) 16, post-natal day (P) 1, P7 [27] and P15 [28], accessed via the gEAR portal (https://umgear.org/). *Klhl18* expression was seen in IHCs, OHCs, pillar cells and Deiters' cells at one or more of these stages. It was not seen at any stage in outer pillar cells, Hensen cells, or inner phalangeal cells (Fig 11). To visualise the expression, we chose 6 marker genes for comparison (*Myo7a* for hair cells, *Fgf8* for inner hair cells, *Slc26a5* for outer hair cells, *Sox2* for non-sensory cells, *S100b* for inner pillar cells and *Hes5* for Deiters' cells) and normalised expression to the housekeeping gene *Hprt*. *Klhl18* is expressed in both inner and outer hair cells at low but increasing levels from P1 to P15, and in Deiters' cells from P15, while its expression in inner pillar cells at P1 and P7 is very low. Its temporal and spatial expression pattern most closely match *Myo7a*, although it is expressed at much lower levels.

## Discussion

We have identified a key role for Klhl18 in the maintenance of cochlear function by tracking auditory responses from normal ABR thresholds at 3 weeks old to progressive deterioration of thresholds from 4 weeks onwards in mice with two different mutant alleles of the *Klhl18* gene. Low frequencies were predominantly affected, which is an unusual pattern. Outer hair cell function was normal judging from DPOAEs and CM responses, and the endocochlear potential was normal indicating normal stria vascularis function. Innervation of the organ of Corti appeared normal and IHCs had normal numbers of synapses with afferent neurons in mutants. However, scanning electron microscopy revealed abnormal tapering of the distal ends of IHC stereocilia in apical regions of the cochlea responsive to lower frequency tones, with a more normal appearance of stereocilia in the basal, high frequency regions.

There are relatively few cases where hearing impairment affects mainly the lower frequency ranges, and these have a variety of underlying pathologies. Mice lacking beta-tectorin have a disrupted tectorial membrane resulting in elevated CAP thresholds at low frequencies [29]. The headbanger mutation of *Myo7a* results in defective stereocilia bundles, resulting in low frequency hearing impairment [30]. Other examples include a low frequency sensorineural hearing impairment in a mouse model of Muenke Syndrome [31] and a low frequency hearing loss of unknown pathology in *Camsap3* mutant mice [1]. In humans, low frequency hearing loss is rare and can be associated with non-syndromic deafness DFNA1 [32], Wolfram Syndrome 1 (WFS1) [33–35] and with Meniere's Disease [36]. The addition of *Klhl18* to this short list represents an important finding.

Stereocilia tapering and disorganised bundles were observed not only at 4 weeks old, where the apical preponderance of tapering correlated with reduced sensitivity to low frequency sounds, but also at 3 weeks old when ABR thresholds were normal in the *Klhl18^lowf^* mutants. Furthermore, less extensive stereocilia tip tapering and hair bundle disorganisation was found in the most apical regions of heterozygotes and wildtypes. This was a surprising finding, reported by all three observers who were blinded to genotype. What might explain this? As the tapering phenotype is seen mainly in the longest row of stereocilia, it may be that sufficient tip-links survive to produce normal IHC signal transduction and hearing function. By 4 weeks

of age, the bundle disruption and tapering effect is more extensive, possibly with the consequence that any remaining tip-links can no longer support normal transduction, causing hearing deterioration. Another potential explanation is that the frequency-place map of the cochlea may be shifted in mice from this colony (in both mutants and wildtypes) allowing hair cells from more basal regions to respond to lower frequencies than usually expected [21]. Our observations suggest that an IHC stereocilia defect is an additional cochlear pathology that can be hidden behind normal ABR thresholds, a phenomenon previously linked to auditory nerve myelination defects [37], to cochlear synaptopathy [38] and to auditory nerve dysfunction [39].

*Klhl18* is a member of the Kelch-like family of proteins, with domains that interact with many other proteins, including interaction of Kelch repeats with actin [40]. Deficiency of Klhl18 may therefore result in a range of cellular problems in highly actin-rich cells such as IHCs, and in particular may affect the actin core of stereocilia leading to the tip tapering phenotype we observed. The tapering stereocilia of IHCs in *Klhl18* mutant mice is similar to that noted in *Espn*[+/je] mutant mice by Sekerková and colleagues [41]. These heterozygote jerker mutant mice showed a transient tapering of apical turn (80% of cochlear length) IHC stereocilia which is apparent at P5 but not at P10. In more extreme apical regions (>95%), this tapering persists until at least 8 months of age. They did not observe tapering in their wildtype mice, but these were CBA/CaJ inbred strain control mice and not littermates of the *Espn*[+/je] mice [41]. Similar to our finding in *Klhl18* mutants, missing stereocilia from the shortest row in OHCs was reported in the *Espn*[+/je] mice [41], and also in mutations of *Espnl* [42] and Myo15 [43]. Espin is known to be involved in actin bundling and the loss of other actin-interacting proteins can also produce occasional narrowed IHC stereocilia, for example plastin1 [44] and Eps8L2 [45].

The related family member Klhl19 is known to co-localize with Myo7a in IHCs in the inner ear [46]. Mutations in *Myo7a* are associated with stereocilia defects [47] including tapering of distal parts of IHC stereocilia [48] and low frequency hearing impairment [30]. If, like Klhl19, Klhl18 is able to bind Myo7a, it may play a role in stereocilia stability and maintenance in a similar way. The growth in expression of *Klhl18* in hair cells during development is similar to that of *Myo7a*, supporting this suggestion.

The observation of tapering stereocilia even in wildtypes is unusual and may indicate that there is another genetic variant in this colony that can also influence stereocilia formation. The mice in this study were all homozygous for a targeted mutation of *Mab21l4* (also known as *2310007B03Rik*) but this mutation alone does not cause increased ABR thresholds [1]. Other mutations may have occurred spontaneously and become fixed within the current colony. Whatever any role of the *Mab21l4* mutation might be, the *Klhl18*[lowf] mutation must exacerbate the tapering phenotype as homozygotes are the most severely affected.

We found abnormalities in the SP measured from the round window in response to low frequency tones. The SP is a complex potential reflecting hair cell depolarisation, with contributions from both OHCs and IHCs [20,25,49]. However, as all other observations of OHCs in this study suggest they are functioning normally in the *Klhl18* mutant mice, the abnormalities noted in the SP are most likely due to IHC dysfunction. The hair bundle defect may impair the IHC receptor potential, disrupting normal voltage changes in the IHC that are required for normal activation of $Ca^{2+}$ channels [50] and therefore synaptic transmission [51].

Auditory Neuropathy Spectrum Disorder (ANSD) is a complex and heterogeneous condition in humans (e.g. [52]). ANSD can be diagnosed by normal otoacoustic emissions along with abnormal ABRs, but there are a wide range of underlying causes, genetic or otherwise (e.g. [53]). Our findings suggest that the *KLHL18* gene will be a good candidate for involvement in human ANSD. Furthermore, studies of the hearing impairment associated with

mutation of *Klhl18* will contribute to a wider understanding of pathophysiological mechanisms contributing to low frequency hearing impairment.

## Supporting information

**S1 Fig. Recordings of Auditory Brainstem Responses (ABRs) in mice. A**, Representative ABRs recorded in response to 12 kHz tone pips presented from 0–85 dB SPL are plotted for a *Klhl18*[+/lowf] control mouse (black) and a *Klhl18*[lowf/lowf] mutant mouse (red). P1 and N1 indicate the positions of the first positive and negative peaks of the waveform, from which amplitude and latency measurements are make. The thickened line on each stack of ABR traces indicates the visually-determined threshold level for these responses (15 dB SPL for the control mouse; 60 dB SPL for the mutant mouse). **B**, Input-Output functions for ABR wave 1 amplitude and P1 latency measurements (from the traces shown in A) are plotted as a function of stimulus sound pressure level (dB SPL), for the control mouse (black circles) and mutant mouse (red up-triangles). Sub-threshold points are indicated by open symbols. Supra-threshold points are indicated by filled symbols. The estimated threshold for each mouse is indicated by the appropriate single symbol on the abscissa of the wave 1 amplitude function.
(TIF)

**S2 Fig. Recordings of distortion product otoacoustic emissions (DPOAEs) in mice. A**. A representative DPOAE frequency spectrum is plotted for a response recorded with a stimulus of 2 frequency components, f1 (10,000 Hz, 70 dB SPL) and f2 (12,000 Hz, 60 dB SPL), for a wildtype *Klhl18*[+/+] mouse. The spectral peaks of the two stimulus tones are labelled f1 and f2. The measured DPOAE is labelled 2f1-f2. These peaks are further highlighted by filled circles. Similar spectra were recorded using f2 levels from -10 dB to 65 dB and used to create a growth function for the 2f1-f2 DPOAE. **B**. For the same *Klhl18*[+/+] mouse, DPOAE amplitude is plotted as a function of f2 level (circles). The thick grey line indicates the mean noise-floor amplitude for each response. The thin grey line indicates 2 standard deviations (SD) above the mean for the noise-floor. This line is used to estimate a threshold for the DPOAE component; the lowest f2 stimulus level where the DPOAE is greater than 2 SDs above the noise-floor. Sub-threshold DPOAEs are indicated by open circles. Supra-threshold DPOAEs are indicated by black filled circles. The estimated threshold is indicated by the single symbol on the abscissa.
(TIF)

**S3 Fig. Round Window Response (RWR), analyses and parameters measured. A.** A typical electrophysiological response recorded from the mouse round window with a 1 Hz-50 kHz bandwidth bioamplifier; in this case, in response to an 18 kHz tone pip at 90 dB SPL. **Bi.** The response in (A), low pass filtered at 3 kHz to remove the cochlear microphonic (CM) component and expose the summating potential (SP) and cochlear compound action potential (CAP). The "X" symbols indicate points on the response at 19 ms and 24 ms. Data points between these time values were averaged to generate the magnitude of the SP. **Bii** illustrates that the SP is measured as a voltage magnitude displaced from the zero baseline of the lowpass filtered response. The SP can be positive or negative relative to the baseline. **Biii.** SP is plotted against dB SPL to produce a SP input-out function (IOF). **Ci.** The RWR from (A), bandpass filtered (300 Hz– 3 kHz) to remove both the CM and SP components and isolate the CAP response, composed of a large negative peak (N) followed by a large positive peak (P), indicated by open circle symbols. **Cii** illustrates the same filtered response from Ci but with the abscissa expanded to clearly show the structure of the CAP waveform. Peaks N and P are again illustrated by open circles. The arrow indicates the peak-to-peak (N to P) magnitude of the CAP. The latency of the N and P peaks was also measured (corrected to account for the 5ms

onset delay of the stimulus tone pip). **Ciii.** The latency of peaks N and P (filled and open circles respectively) are plotted against dB SPL to produce a latency IOF. **Civ.** Similarly, the amplitude of peaks N and P and the N-P amplitude are plotted against dB SPL to produce amplitude IOFs. **Di.** The RWR from (A) bandpass filtered, centred on the stimulus frequency with high- and low- pass corner frequencies at ± 100 Hz, to remove all components within the response other than the CM. **Dii.** The filtered CM response is limited to a 7–23 ms time window before applying a Fast Fourier Transformation; the resulting power spectrum is shown with the magnitude of the CM component indicated by an open circle. **Diii**. CM amplitudes measured across all stimulus level presented are plotted against dB SPL to produce a CM IOF.
(TIF)

**S4 Fig. Male vs female responses of 6-weeks old *Klhl18* mice.** Mean (± SD) thresholds for male (blue triangles) and female (pink circles) *Klhl18* control mice (open symbols) and *Klhl18* mutant mice (filled symbols) are plotted in **A** ABR thresholds of *Klhl18$^{lowf}$* mice (n = 6 control male, n = 6 control female, n = 5 mutant male and n = 8 mutant female), **B** ABR thresholds of *Klhl18$^{tm1a}$* mice(n = 6 control female, n = 3 mutant male and n = 3 mutant female), **C** DPOAE thresholds of *Klhl18$^{lowf}$* mice(n = 6 control male, n = 6 control female, n = 5 mutant male and n = 8 mutant female), **D** DPOAE thresholds of *Klhl18$^{tm1a}$* mice(n = 6 control female, n = 3 mutant male and n = 3 mutant female), **E** CAP thresholds of *Klhl18$^{lowf}$* mice (n = 7 wildtype male, n = 3 wildtype female, n = 4 mutant male and n = 6 mutant female) and **F** CM thresholds of *Klhl18$^{lowf}$* mice (n = 7 wildtype male, n = 3 wildtype female, n = 4 mutant male and n = 6 mutant female). Where wildtype and heterozygote mice were combined to form a control group, these groups consisted of (A) 1 wildtype male, 5 heterozygote males, 2 wildtype females, 4 heterozygote females, (B) 1 wildtype female, 5 heterozygote females, (C) 1 wildtype male, 5 heterozygote males, 2 wildtype females, 4 heterozygote females and (D) 1 wildtype female, 5 heterozygote females. Results of a mixed-effects model statistical analysis between male and female control and mutant thresholds are shown in S1 Table. Sidak's multiple comparisons test was used to examine the difference between control and mutant thresholds for each stimulus. Significant differences are indicated here by *.
(TIF)

**S5 Fig. Age-related changes in ABR thresholds of *Klhl18$^{tm1a}$* mice.** A summary of ABR thresholds estimated from control (wildtype or heterozygous) and mutant (homozygous) are shown for *Klhl18$^{tm1a}$* mice aged 2 weeks (**A**, n = 9 control, including 3 wildtype, n = 9 mutant), 3 weeks (**B**, n = 4 control, all wildtype, n = 7 mutant), 4 weeks (**C**, n = 8 control, including 3 wildtype, n = 8 mutant), 6 weeks (**D**, n = 6 control, including 1 wildtype, n = 6 mutant), 8 weeks (**E**, n = 8 control, including 1 wildtype, n = 8 mutant) and 14 weeks (**F**, n = 8 control, including 3 wildtype, n = 8 mutant). Thresholds from individual mutant mice are indicated by grey circles. Mean threshold (±SD) are plotted for control mice as black down-triangles and for mutant mice as red up-triangles. Ck: Click stimulus. Results of a mixed-effects model statistical analysis between control and mutant thresholds are shown in S1 Table. Sidak's multiple comparisons test was used to examine the difference between control and mutant thresholds for each stimulus. Significant differences are indicated here by *.
(TIF)

**S6 Fig. Mean ABR waveforms of *Klhl18$^{tm1a}$* mice at 65dB SPL and 20 dB SL.** Group averaged ABR waveforms, evoked by 65 dB SPL stimuli, are plotted for mutant mice and heterozygous littermate controls aged 2 weeks (**A-C**), 3 weeks (**D-F**) and 6 weeks (**G-I**) in response to click stimuli (**A,D,G**), 12 kHz stimuli (**B,E,H**) and 24 kHz stimuli (**C,F,I**). Group averaged ABR waveforms, evoked by stimuli presented at 20 dB above threshold (20 dB sensation level,

dB SL), are plotted for mice aged 2 weeks (**J-L**), 3 weeks (**M-O**), 6 weeks (**P-R**) in response to click stimuli (**J,M,P**), 12 kHz stimuli (**K,N,Q**) and 24 kHz stimuli (**L,O,R**). Group sizes were, at 2 weeks old, n = 9 control mice (n = 3 wildtypes and n = 6 heterozygotes), n = 9 mutants; at 3 weeks old, n = 4 control mice (all wildtypes), n = 7 mutants; at 6 weeks old, n = 6 controls (n = 1 wildtype and n = 5 heterozygotes), n = 6 mutants. Mean amplitude waveforms are plotted for the control mice (black lines) and mutant mice (red lines).
(TIF)

**S7 Fig. ABR wave 1 amplitude and latency input-output functions in *Klhl18^tm1a^* mice.**
Mean ABR wave 1 P1-N1 peak-to-peak amplitude (±SD) as a function of stimulus level is plotted for mice aged 2, 3 and 6 weeks old are plotted in **A-C** (n = 9 controls, including 3 wildtypes; n = 9 mutants), **D-F** (n = 4 wildtype controls, n = 7 mutants) and **G-I** (n = 6 controls, including 1 wildtype; n = 6 mutants), respectively. Results are plotted for click stimuli (**A,D,G**), 12 kHz tones (**B,E,H**) and 24 kHz tones (**C,F,I**). Data from control mice are plotted as black down-triangles. Data from mutant mice are plotted as red up-triangles. Mean ABR wave P1 latency (±SD) as a function of stimulus level is plotted for the same mice aged 2, 3 and 6 weeks old are plotted in **J-L**, **M-O** and **P-R**, respectively. Results are plotted for click stimuli (**J,M,P**), 12 kHz tones (**K,N,Q**) and 24 kHz tones (**L,O,R**). Results of a mixed-effects model statistical analysis between control and mutant thresholds are shown in S1 Table. Sidak's multiple comparisons test was used to examine the difference between control and mutant data for each stimulus. Significant differences are indicated here by *.
(TIF)

**S8 Fig. ABR wave 1 amplitude and latency input-output functions in *Klhl18^tm1a^* heterozygote mice.** Mean ABR wave 1 P1-N1 peak-to-peak amplitude (±SD) as a function of stimulus level is plotted for mice aged 2 and 6 weeks old in **A-C** (n = 3 wildtype; n = 6 heterozygotes), and **D-F** (n = 1 wildtype; n = 5 heterozygotes), respectively. Results are plotted for click stimuli (**A,D**), 12 kHz tones (**B,E**) and 24 kHz tones (**C,F**). Data from wildtype mice are plotted as black down-triangles. Data from heterozygote mice are plotted as open blue up-triangles. Mean ABR wave P1 latency (±SD) as a function of stimulus level is plotted for the same mice aged 2 and 6 weeks old are plotted in **G-I** and **J-L**, respectively. Results are plotted for click stimuli (**G,J**), 12 kHz tones (**H,K**) and 24 kHz tones (**I,L**). Results of a mixed-effects model statistical analysis between control and mutant thresholds are shown in S1 Table. Sidak's multiple comparisons test was used to examine the difference between wildtype and heterozygous mouse data for each stimulus. No significant differences were observed.
(TIF)

**S9 Fig. DPOAE growth and thresholds in *Klhl18^tm1a^* mice.** DPOAE results from mutant mice and littermate controls aged 2, 3 and 6 weeks old are plotted in **A-F** (n = 5 controls, comprised of 2 wildtypes and 3 heterozygotes; n = 5 mutants), **G-L** (n = 4 wildtype controls; n = 7 mutants) and **M-N** (n = 6 controls, comprised of 1 wildtype and 5 heterozygotes; n = 6 mutants), respectively. Data from control mice are plotted as black down-triangles. Data from mutant mice are plotted as red up-triangles. The mean (±SD) amplitude of the 2f1-f2 DPOAE (dB SPL) is plotted as a function of f2 stimulus level (dB SPL) for f2 frequencies of 6 kHz (**A, G, M**), 12 kHz (**B, H, N**), 18 kHz (**C, I, O**), 24 kHz (**D, J, P**) and 30 kHz (**E, K, Q**). Mean threshold (±SD) of the 2f1-f2 DPOAE (derived from individual growth functions, eg shown in S2 Fig) and plotted in **F**, **L** and **R** for mice aged 2 weeks, 3 weeks and 6 weeks respectively. In addition to the mean data, thresholds from individual mutant mice are plotted as grey open circles. Results of a mixed-effects model statistical analysis between control and mutant data are shown in S1 Table. Sidak's multiple comparisons test was used to examine the difference

between control and mutant data for each stimulus. Significant differences are indicated here by *.
(TIF)

**S10 Fig. RWRs and components in a wildtype control mouse.** Representative responses recorded for 18 kHz tones in a control mouse, over stimulus levels from 0–90 dB SPL are illustrated. **A.** Broadband RWRs (1 Hz– 50 kHz). **B.** Lowpass filtered (3 kHz) RWRs show responses containing the CAP and SP. **C.** Bandpass filtered (300 Hz—3 kHz) RWRs show responses containing the CAPs. **D.** Bandpass filtered (18 kHz ± 100 Hz) RWRs show responses containing the CM.
(TIF)

**S11 Fig. RWRs and components in a *Klhl18^{lowf}* mutant mouse.** Representative responses recorded for 18kHz tones in a control mouse, over stimulus levels from 0–90 dB SPL are illustrated. **A.** Broadband RWRs (1 Hz– 50 kHz). **B.** Lowpass filtered (3 kHz) RWRs show responses containing the CAP and SP. **C.** Bandpass filtered (300 Hz– 3 kHz) RWRs show responses containing the CAPs. **D.** Bandpass filtered (18 kHz ± 100 Hz) RWRs show responses containing the CM.
(TIF)

**S12 Fig. Round Window Response measurements from *Klhl18^{+/lowf}* (heterozygote) mice.** Compound Action Potential latency (of wave N) and amplitude (N-P amplitude) are plotted in **A**-**E** and **F**-**J**, respectively, for potentials measured in response to tones of 6, 12, 18, 24 and 30 kHz. Data are plotted as mean ± SD for wildtype control mice (n = 10; black down-triangles) and heterozygote mice (n = 7; blue up-triangles). **K.** Mean ± SD of the CAP threshold is plotted against stimulus frequency for wildtype control mice (black down-triangles) and heterozygote mice (blue up-triangles). Open circles and grey lines indicate CAP thresholds for individual heterozygote mice. **L**-**P** plot mean (± SD) Cochlear Microphonic amplitude (dB re 1 μV) against stimulus level (dB SPL) for wildtype mice (black down-triangles) and heterozygote mice (blue up-triangles). The line plotted in grey represents the mean ± SD amplitude of stimulus artifact. **Q.** Mean ± SD of the estimated CM threshold is plotted against stimulus frequency for wildtype control mice (black down-triangles) and heterozygote mice (blue up-triangles). Open circles and grey lines indicate CM thresholds for individual heterozygote mice. The grey line without symbols indicates the estimated magnitude of the stimulus artifact, as described in Fig 6. **Ri**-**Vi** plot mean (± SD) Summating Potential amplitude (μV) against stimulus level (dB SPL) for wildtype mice (black down-triangles) and heterozygote mice (blue up-triangles). **Rii**-**Vii** plot SP IOFs for individual control (black lines) and mutant (blue lines) mice.
(TIF)

**S1 Table. Statistical test results.**
(XLSX)

# Acknowledgments

We thank the Wellcome Sanger Institute Mouse Genetics Project for access to the *Klhl18^{tm1a (KOMP)Wtsi}* and *Mab21l4^{tm1a(KOMP)Wtsi}* mutant mice; Dr Elisa Martelletti for advice on confocal microscopy; Dr Selina A. Pearson for ABR recordings to help to establish the *Klhl18^{lowf}* colony; and Dr Douglas C. Fitzpatrick for helpful suggestions on round window recordings. For the purpose of Open Access, the author has applied a CC BY public copyright licence to any Author Accepted Manuscript version arising from this submission.

## Author Contributions

**Conceptualization:** Neil J. Ingham, Karen P. Steel.

**Data curation:** Neil J. Ingham.

**Formal analysis:** Neil J. Ingham, Navid Banafshe, Jing Chen, Morag A. Lewis.

**Funding acquisition:** Karen P. Steel.

**Investigation:** Neil J. Ingham, Navid Banafshe, Clarisse Panganiban, Julia L. Crunden, Morag A. Lewis.

**Methodology:** Neil J. Ingham, Karen P. Steel.

**Project administration:** Neil J. Ingham, Karen P. Steel.

**Resources:** Karen P. Steel.

**Software:** Neil J. Ingham.

**Supervision:** Karen P. Steel.

**Visualization:** Neil J. Ingham, Navid Banafshe, Julia L. Crunden, Jing Chen, Morag A. Lewis.

**Writing – original draft:** Neil J. Ingham.

**Writing – review & editing:** Neil J. Ingham, Navid Banafshe, Clarisse Panganiban, Julia L. Crunden, Jing Chen, Morag A. Lewis, Karen P. Steel.

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
