## [Decision Letter · Decision Letter 0]

3 Sep 2021

PONE-D-21-24337

Inner hair cell dysfunction in Klhl18 mutant mice leads to low frequency progressive hearing loss

PLOS ONE

Dear Dr. Ingham,

Thank you for submitting your manuscript to PLOS ONE. After careful consideration, we feel that it has merit but does not fully meet PLOS ONE’s publication criteria as it currently stands. Therefore, we invite you to submit a revised version of the manuscript that addresses the points raised during the review process.

The manuscript needs the minor revisions suggested by Reviewer 1.

We look forward to receiving your revised manuscript.

Kind regards,

Olivia Bermingham-McDonogh, Ph.D.

Academic Editor

PLOS ONE

Journal Requirements:

3. Please include a copy of Table S1 which you refer to in your text on page 13, 14, 16, 17, 18, 20, 21, 38, 39, 40 and 41.

Reviewers' comments:

Reviewer's Responses to Questions

**Comments to the Author**

1. Is the manuscript technically sound, and do the data support the conclusions?

Reviewer #1: Yes

Reviewer #2: Yes

2. Has the statistical analysis been performed appropriately and rigorously? 

Reviewer #1: Yes

Reviewer #2: Yes

3. Have the authors made all data underlying the findings in their manuscript fully available?

Reviewer #1: Yes

Reviewer #2: Yes

4. Is the manuscript presented in an intelligible fashion and written in standard English?

Reviewer #1: Yes

Reviewer #2: Yes

5. Review Comments to the Author

Reviewer #1: The manuscript from Neil Ingham and co-authors entitled “Inner hair cell dysfunction in Klhl18 mutant mice leads to low frequency progressive hearing loss” is precisely crafted and has an abundance of supporting data that is presented in good-looking figures. It so nice at least once in a while to read a manuscript in near final form with thorough evaluations and properly replicated data.

There are only a few minor issues to suggest to the authors.

In the abstract and text there are sentences such as “Scanning electron microscopy showed that Klhl18lowf mutant mice had abnormally tapering inner hair cell stereocilia in the apical half of the cochlea while their synapses appeared normal.” The stereocilia taper usually refers to the base of a stereocilium where the stereocilium joins with the apical surface of the hair cell. That’s not the taper the authors are referring to and that becomes obvious when examining the beautiful images provided. The authors should make an effort to better define both in the abstract and in the main text that they are referring to abnormal tapers near the distal end of stereocilia.

The first time a variant of Klhl18 is mentioned, indicate if the allele is dominant or recessive.

The statement on line 51 “Neither allele produced overt vestibular phenotypes and homozygotes are viable and fertile [1,2].” could be improved by mentioning the vestibular tests. “overt” isn’t very scientific. What vestibular abnormal phenotype would be classified as overt that these mice didn’t exhibit?

Reviewer #2: Inner hair cell dysfunction in Klhl18 mutant mice leads to low frequency progressive hearing loss

The manuscript deals with characterization of a model for a hearing phenotype that is found in, but less well studied, in the human population. The Klhl18 (Kelch-like family member 18) gene was examined using two different alleles in the mouse. Interestingly, this gene has been implicated in retina function. The morphological changes in the mice were subtle, with normal DPOAE, CM and synaptic contacts. However, the inner hair cells had some morphological changes (tapering in the inner hair cells), backed up by summating potentials, implicating these cells as impaired in the Klhl18 mutants. The analysis is extremely thorough, and provides a mechanism of function for the low frequency progressive hearing loss due to mutations in this gene. It will be interesting to see if human pathogenic variants might turn up if such families will be screened by whole exome sequencing.

6. PLOS authors have the option to publish the peer review history of their article (what does this mean?). If published, this will include your full peer review and any attached files.

Reviewer #1: No

Reviewer #2: No

---

## [Author Response · Author response to Decision Letter 0]

17 Sep 2021

Responses to reviewers’ comments, Ingham et al.

1. When submitting your revision, we need you to address these additional requirements. Please ensure that your manuscript meets PLOS ONE's style requirements, including those for file naming. The PLOS ONE style templates can be found at https://journals.plos.org/plosone/s/file?id=wjVg/PLOSOne_formatting_sample_main_body.pdf and https://journals.plos.org/plosone/s/file?id=ba62/PLOSOne_formatting_sample_title_authors_affiliations.pdf

>> We have edited the manuscript to comply with the journal formatting requirements and tracked all changes.

>> We have checked the reference list but have not noticed any publications that we know have been retracted.

3. Please include a copy of Table S1 which you refer to in your text on page 13, 14, 16, 17, 18, 20, 21, 38, 39, 40 and 41.

>> We apologize for overlooking the inclusion of Table S1 in the original submission. An Excel file containing Table S1 (named “_TableS1_Ingham et al.xlsx”) is included with this revision. 

Reviewer #1: The manuscript from Neil Ingham and co-authors entitled “Inner hair cell dysfunction in Klhl18 mutant mice leads to low frequency progressive hearing loss” is precisely crafted and has an abundance of supporting data that is presented in good-looking figures. It so nice at least once in a while to read a manuscript in near final form with thorough evaluations and properly replicated data.

>>Thank you for these comments.

There are only a few minor issues to suggest to the authors.

Klhl18lowf mutant mice had abnormally tapering inner hair cell stereocilia in the apical half of the cochlea while their synapses appeared normal.” The stereocilia taper usually refers to the base of a stereocilium where the stereocilium joins with the apical surface of the hair cell. That’s not the taper the authors are referring to and that becomes obvious when examining the beautiful images provided. The authors should make an effort to better define both in the abstract and in the main text that they are referring to abnormal tapers near the distal end of stereocilia.

>> Thank you for pointing out that confusion. We have altered the text at multiple locations to clarify that we are talking about tapering of the tips of stereocilia, not the insertion point into the cuticular plate.

The first time a variant of Klhl18 is mentioned, indicate if the allele is dominant or recessive.

>> We have added comments in the abstract and introduction that the hearing loss is recessive in inheritance. However, there may be a semi-dominant effect on stereocilia tip tapering. It is the phenotype studied that is recessive or dominant rather than the allele, so we are cautious in the text to describe hearing loss as recessive, not the full phenotype. We found no differences in our measures of auditory function between heterozygotes and wildtypes.

The statement on line 51 “Neither allele produced overt vestibular phenotypes and homozygotes are viable and fertile [1,2].” could be improved by mentioning the vestibular tests. “overt” isn’t very scientific. What vestibular abnormal phenotype would be classified as overt that these mice didn’t exhibit?

>> We have added “such as circling or head-bobbing” to the text at this point to clarify.

Reviewer #2: Inner hair cell dysfunction in Klhl18 mutant mice leads to low frequency progressive hearing loss

The manuscript deals with characterization of a model for a hearing phenotype that is found in, but less well studied, in the human population. The Klhl18 (Kelch-like family member 18) gene was examined using two different alleles in the mouse. Interestingly, this gene has been implicated in retina function. The morphological changes in the mice were subtle, with normal DPOAE, CM and synaptic contacts. However, the inner hair cells had some morphological changes (tapering in the inner hair cells), backed up by summating potentials, implicating these cells as impaired in the Klhl18 mutants. The analysis is extremely thorough, and provides a mechanism of function for the low frequency progressive hearing loss due to mutations in this gene. It will be interesting to see if human pathogenic variants might turn up if such families will be screened by whole exome sequencing.

>> Thank you for these comments. We are looking out for human mutations as the reviewer suggests.

---

## [Editor Report · Decision Letter 1]

20 Sep 2021

Inner hair cell dysfunction in Klhl18 mutant mice leads to low frequency progressive hearing loss

PONE-D-21-24337R1

Dear Dr. Ingham,

We’re pleased to inform you that your manuscript has been judged scientifically suitable for publication and will be formally accepted for publication once it meets all outstanding technical requirements.

Kind regards,

Olivia Bermingham-McDonogh, Ph.D.

Academic Editor

PLOS ONE
---

## [Editor Report · Acceptance letter]

24 Sep 2021

PONE-D-21-24337R1 

Inner hair cell dysfunction in *Klhl18* mutant mice leads to low frequency progressive hearing loss 

Dear Dr. Ingham:

I'm pleased to inform you that your manuscript has been deemed suitable for publication in PLOS ONE. Congratulations! Your manuscript is now with our production department. 

Kind regards, 

on behalf of

Dr. Olivia Bermingham-McDonogh 

Academic Editor

PLOS ONE